# *Frizzled3* controls axonal development in distinct populations of cranial and spinal motor neurons

**Zhong L Hua[1], Philip M Smallwood[1,2], Jeremy Nathans[1,2,3,4]\***

[1]Department of Molecular Biology and Genetics, Johns Hopkins University School of Medicine, Baltimore, United States; [2]Howard Hughes Medical Institute, Johns Hopkins University School of Medicine, Baltimore, United States; [3]Department of Neuroscience, Johns Hopkins University School of Medicine, Baltimore, United States; [4]Department of Ophthalmology, Johns Hopkins University School of Medicine, Baltimore, United States

**Abstract** Disruption of the *Frizzled3* (*Fz3*) gene leads to defects in axonal growth in the VII[th] and XII[th] cranial motor nerves, the phrenic nerve, and the dorsal motor nerve in fore- and hindlimbs. In *Fz3*[−/−] limbs, dorsal axons stall at a precise location in the nerve plexus, and, in contrast to the phenotypes of several other axon path-finding mutants, *Fz3*[−/−] dorsal axons do not reroute to other trajectories. Affected motor neurons undergo cell death 2 days prior to the normal wave of developmental cell death that coincides with innervation of muscle targets, providing in vivo evidence for the idea that developing neurons with long-range axons are programmed to die unless their axons arrive at intermediate targets on schedule. These experiments implicate planar cell polarity (PCP) signaling in motor axon growth and they highlight the question of how PCP proteins, which form cell–cell complexes in epithelia, function in the dynamic context of axonal growth.

**\*For correspondence:** jnathans@jhmi.edu

**Reviewing editor**: Robb Krumlauf, Stowers Institute for Medical Research, United States

## Introduction

The vertebrate nervous system is highly evolved to extract, integrate, retrieve, and transform information, a functional sophistication that is mirrored in the complexity of its underlying neural circuits. These circuits are produced by precisely orchestrated patterns of neuronal proliferation, migration, and differentiation, and by the establishment of specific synaptic connections between neurons and their targets. For neurons with distant targets, making the appropriate connections depends on accurate long-range navigation by their growing axons (*Dickson, 2002*; *O'Donnell et al., 2009*).

Motor neurons represent an attractive system in which to study axonal growth and guidance because motor axons generally grow over great distances along highly stereotyped pathways. Motor neurons in the brainstem extend their axons within cranial nerves to control muscles involved in feeding, facial expression, and eye, neck, and head movements (*Guthrie, 2007*). Motor neurons in the spinal cord are arranged in longitudinal columns along the rostrocaudal axis, and the neurons in different columns innervate distinct peripheral targets (*Landmesser, 1978*; *Bonanomi and Pfaff, 2010*). Spinal motor neurons in the lateral motor column (LMC) innervate the limbs and are therefore present only at brachial and lumbar levels. The LMC is divided into lateral (LMC$_L$) and medial (LMC$_M$) divisions, which innervate, respectively, the dorsal and ventral limb musculature.

For motor axons that innervate the limb musculature, different signaling systems—including the Cxcl12/Cxcr4, GDNF/Ret, Ephrin/Eph, and Semaphorin/Neuropilin (Npn) systems—provide information at different decision points during axonal navigation (*Helmbacher et al., 2000*; *Huber et al., 2005*; *Lieberam et al., 2005*; *Kramer et al., 2006*; *Luria et al., 2008*; *Bonanomi et al., 2012*; reviewed in

**eLife digest** For the nervous system to become wired up correctly, neurons within the developing embryo must project over long distances to form connections with remote targets. They do this by lengthening their axons—the 'cables' along which electrical signals flow—and some axons in adult humans can grow to be more than 1 metre long.

This type of long-range pathfinding activity is particularly common for neurons that control movement, as many of these neurons must establish connections with muscles that are some distance away from the brain. For example, motor neurons in the brainstem form connections with muscles in the face to control facial expressions, while motor neurons in parts of the spinal cord project to muscles in the limbs. Multiple signaling pathways tell the developing axons which direction to grow en route to their final targets.

Now, Hua et al. have shown that an evolutionarily conserved protein called Frizzled3 is also involved in this process. In mouse embryos that lacked Frizzled3, the motor nerves that control breathing and limb movements were thinner than those in normal mice. In the mutant animals, many motor axons failed to form connections with their targets. Instead, these axons came to an abrupt halt midway along their intended paths and the neurons from which they originated died soon afterwards. These experiments support the idea that developing neurons are programmed to die unless their axons progress on the appropriate schedule.

As well as increasing our knowledge of the networks of connections that form within the developing mammalian nervous system, the work of Hua et al. provides new insights into some of the molecular mechanisms by which these connections are established.

*Bonanomi and Pfaff, 2010*). Cxcl12/Cxcr4 signaling is required for the initial emergence of motor axons from the ventral face of the spinal cord, with ventral axons expressing the Cxcr4 receptor and thereby conferring responsiveness to Cxcl12, which is produced by the adjacent mesenchyme. Sema3A, which is produced throughout the limb bud mesenchyme, acts on Npn1-expressing motor axons shortly after these axons emerge from the spinal cord to control the timing of outgrowth and axon fasciculation; and Sema3F, which is produced in the dorsal limb mesenchyme, repels Npn2-expressing $LMC_M$ axons to promote a ventral trajectory. The dorsal/ventral decision is also controlled by cooperative and mutually reinforcing interactions between Ephrin/Eph and GDNF/Ret signaling. GDNF, which is produced in the dorsal limb bud, attracts $LMC_L$ axons, which express high levels of the Ret receptor. Ephrin-A, which is produced in the ventral limb bud, repels $LMC_L$ axons, which express EphA4. At present, the signaling pathways that determine more distal axon branching patterns and control the specificity of nerve-muscle recognition remain largely unknown.

The experiments described here reveal an important role for Frizzled signaling in motor axon growth. Frizzled receptors are found throughout the animal kingdom and mediate at least three distinct types of signaling: (1) canonical signaling, in which Wnt binding to Frizzled activates an Lrp co-receptor to elicit stabilization of cytosolic beta-catenin, which then migrates to the nucleus to control gene expression in cooperation with LEF/TCF transcription factors; (2) tissue polarity or planar cell polarity (PCP) signaling, which acts in many epithelia and features the asymmetric localization of Frizzled and several other proteins in plasma membrane complexes to control cytoskeletal organization; and (3) Wnt/calcium signaling, the least understood of the three pathways, in which Wnt binding to Frizzled receptors leads to release of intracellular calcium (*Angers and Moon, 2009*; *Nusse, 2012*). The PCP pathway was discovered in *Drosophila* as a regulator of cuticular hair and bristle polarity (*Simons and Mlodzik, 2008*). The core set of PCP proteins defined by the *Drosophila* experiments—Frizzled, Dishevelled, Van Gogh/Strabismus, Prickle, Flamingo/Starry Night, and Diego—are conserved in vertebrates. Over the past 15 years, studies in mice, frogs, and zebrafish have shown that core PCP genes play central roles in a wide variety of developmental processes that involve directional information: coordinating cell movements during neural tube elongation and closure (convergent/extension movements), setting up the global orientation of hairs within mammalian skin, orienting motile cilia within epithelia to produce coherent fluid flow, and orienting asymmetric bundles of stereocilia on the apical faces of auditory and vestibular sensory hair cells (*Wang and Nathans, 2007*; *Bayly and Axelrod, 2011*; *Goodrich and Strutt, 2011*).

In mammals, two of the ten Frizzled family members—Frizzled3 (Fz3) and Fz6—appear to be devoted largely or exclusively to tissue polarity signaling. Fz6 controls hair orientation, and Fz3 and Fz6 act redundantly to control inner ear sensory hair cell orientation and neural tube closure (*Guo et al., 2004*; *Wang et al., 2006a*). Among Frizzled receptors, Fz3 is unique in its role in controlling axon guidance in the central nervous system (CNS), where it is required for the development of major fiber tracts, including the corpus callosum, and the thalamocortical, corticothalamic, and nigrostriatal tracts (*Wang et al., 2002*, *2006b*). Fz3 appears to be part of a system that provides directional information to growing axons, as indicated by the observation that in *Fz3*$^{-/-}$ embryos commissural sensory axons in the spinal cord orient randomly instead of turning rostrally after midline crossing (*Lyuksyutova et al., 2003*). Virtually identical CNS axon phenotypes are observed in embryos lacking Celsr3, one of three mammalian homologues of the core PCP protein Flamingo/Starry Night (*Tissir et al., 2005*). Fz3 and Celsr1-Celsr3 are also required for caudal and tangential migration of VII[th] cranial (facial branchiomotor) nerve neurons within the developing brainstem (*Qu et al., 2010*). In the periphery, Fz3 and Celsr3 are required for correctly orienting enteric neurons within the gastrointestinal tract, and Fz3 is required for the development of sympathetic chain ganglia and innervation of sympathetic targets (*Armstrong et al. 2011*; *Sasselli et al., 2013*).

In the present work, we have identified a hitherto unrecognized role for Fz3 in the growth of motor axons. Using conventional and conditional loss-of-function alleles for *Fz3*, we show that *Fz3* is required for a subset of cranial and spinal motor axons to grow to their peripheral targets. In the limb, the absence of *Fz3* leads to a specific stalling defect among LMC$_L$ axons that is distinct from the phenotypes produced by defects in several other signaling systems active in limb axon guidance. Affected cranial and spinal motor neurons die shortly after their axons stall, providing in vivo evidence for the general idea that as growing axons arrive at intermediate targets essential survival signals are relayed to their cell bodies.

## Results

### A survey of peripheral nerve defects in *Fz3*$^{-/-}$ embryos

*Fz3* is widely expressed in the developing mouse CNS, and therefore it seemed possible that its loss might impair neurodevelopmental processes other than those previously described in the forebrain (*Wang et al., 2002*, *2006b*). To systematically assess peripheral axon growth and guidance in E11.5-E13.5 embryos, we visualized Neurofilament-M (hereafter 'NF') immunoreactivity in whole-mount tissues and in thick (0.7–1 mm) vibratome sections. We also used the *Hb9-EGFP* transgene to visualize spinal motor axons, including the phrenic nerve, and the motor component of a subset of cranial nerves (*Wichterle et al., 2002*; *Huettl and Huber, 2011*).

In E11.5-E13.5 *Fz3*$^{-/-}$ embryos anti-NF immunostaining revealed a thinning of the XII[th] (hypoglossal) nerve (*Figure 1B,B'*), the phrenic nerve (*Figure 1C,C'*), and the dorsal nerves of the fore- and hindlimbs (described more fully below). Flatmounts of E18.5 *Hb9-EGFP;Fz3*$^{-/-}$ diaphragms co-stained for GFP and nicotinic acetylcholine receptors (using alpha-bungarotoxin; 'alpha-BTX') showed variability in phrenic nerve thickness within the diaphragm that correlated with nerve branching defects, and a decrease in the number of neuromuscular junctions (*Figure 1D–H''*). The innervation defect was especially common in the most ventral region of the diaphragm where many alpha-BTX binding sites were scattered across the muscle surface, both diffusely and in clusters, indicative of a failure of neuromuscular junction formation.

We have classified phrenic nerve defects as mild (the entire diaphragm innervated, but thin phrenic nerve and a decrease in the number of branches and synapses), moderate (similar to mild defects but, additionally, ventral diaphragm not innervated), severe (similar to moderate defects, and, additionally, the crus of the diaphragm not innervated), and complete (phrenic nerve is very thin with minimal innervation across the diaphragm). Among 23 *Fz3*$^{-/-}$ hemi-diaphragms analyzed, the frequencies of these classes were 4/23, 14/23, 4/23, and 1/23, respectively (*Figure 1E–H''*). Among 42 *wildtype* (*WT*) or *Fz3*$^{+/-}$ control hemi-diaphragms, all showed complete diaphragm innervation, with alpha-BTX binding sites tightly clustered at nerve terminals (*Figure 1D–D''*).

### Diverse defects in cholinergic neurons in *Fz3*$^{-/-}$ embryos

Most peripheral nerves are composed of both motor and sensory axons, and therefore a defect in one of these components would be expected to produce a decrease in nerve thickness but not a complete

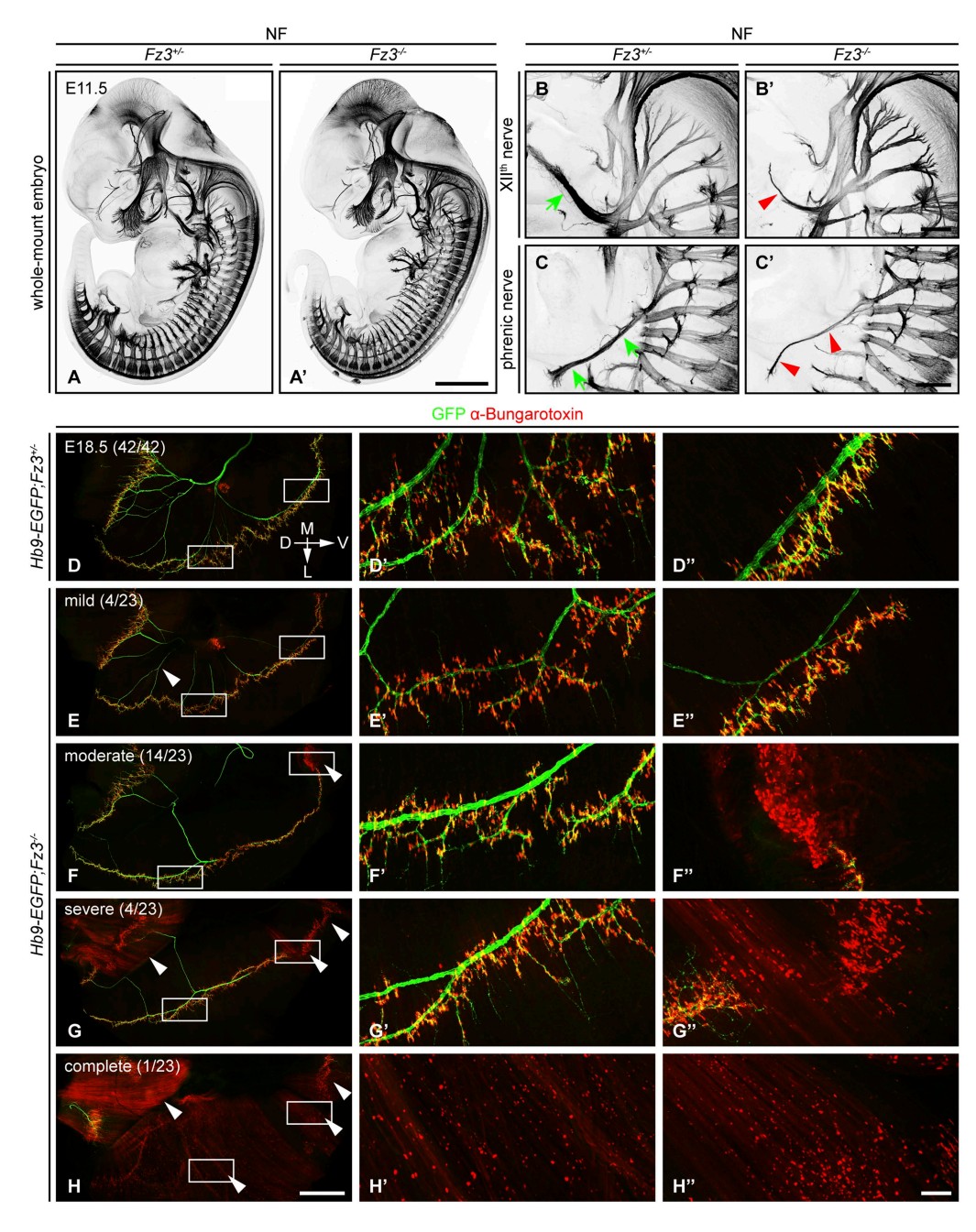

**Figure 1**. Defects in the XII[th] cranial (hypoglossal) and phrenic nerves in *Fz3[−/−]* embryos. (**A–C'**) Anti-NF immunos-taining shows thinned XII[th] and phrenic nerves (green arrows in [**B**] and [**C**] vs red arrowheads in [**B'**] and [**C'**]) in E11.5 *Fz3[−/−]* embryos. The nerve roots and the proximal segment of the XII[th] nerve appear to be unaffected. Enlarged views of the phrenic and XII[th] nerves in (**B–C'**) are maximum intensity projections from consecutive Z stacks. Scale bars: (**A'**), 1 mm; (**B'**) and (**C'**), 200 *µm*. (**D–H''**) Flatmounts of the diaphragm show variably defective motor innervation in E18.5 *Hb9-EGFP;Fz3[−/−]* embryos. Panels (**D–H**) show half of a diaphragm with the midline at the top. (**D'–H'**) and (**D''–H''**) show enlargements of the lateral and ventral regions, respectively, boxed in (**D–H**). D, dorsal; V, ventral; M, medial; L, lateral. (**E–H**) show examples of mild, moderate, severe, and complete phenotypes, with the fraction of embryos in each class shown in parentheses. Scale bars: (**H**), 1 mm; (**H''**), 100 *µm*.

absence of the nerve when visualized with NF immunostaining. To search selectively for defects in cholinergic neurons, including motor neurons, we examined *Fz3⁻/⁻* embryos that carried a *ChAT-IRES-Cre* knock-in allele and a Cre-controlled alkaline phosphatase (AP) reporter, *R26iAP* (*Badea et al., 2009*). In the mouse embryo, *Choline acetyl transferase* (*ChAT*) expression is first detectable at ~E13.5 (*Schambra et al., 1989*; *Ibáñez et al., 1991*). However, the rate of accumulation of endogenous ChAT protein is insufficient for robust immunostaining of axons in the prenatal period. By contrast, in *ChAT-IRES-Cre*;*R26iAP* mice, Cre-mediated activation of the *AP* reporter leads to sufficient AP accumulation by E18.5 for histochemical labeling of both cholinergic cell bodies and axons. Since *Fz3⁻/⁻* mice die as neonates, the high sensitivity of the *ChAT-IRES-Cre*;*R26iAP* method recommended it for surveying late prenatal cholinergic axons.

Cholinergic neurons in the mouse telencephalon can be divided into two major groups: local interneurons in the striatum and projection neurons in the basal forebrain. As seen in *Figure 2B*, striatal cholinergic neurons are present in E18.5 *ChAT-IRES-Cre*;*R26iAP*;*Fz3⁻/⁻* forebrains, but a major cholinergic fiber tract passing through the striatum is missing (compare *Figure 2Bf,Bg* vs *Figure 2Bf',Bg'*). In *Fz3⁻/⁻* brains, cortical and thalamic axons fail to enter the internal capsule and, as a result, the cerebral cortex and the thalamus are disconnected (*Wang et al., 2002*, *2006b*). The cholinergic fiber tract that is missing in the E18.5 *Fz3⁻/⁻* forebrain could depend on the same internal capsule navigation system that instructs cortical and thalamic fibers (*Zhou et al., 2008*).

AP-stained serial sections from *ChAT-IRES-Cre*;*R26iAP*;*Fz3⁺/⁻* and *ChAT-IRES-Cre*;*R26iAP*;*Fz3⁻/⁻* embryos at E18.5 revealed a diverse array of defects among cholinergic neurons and fibers in the brainstem and periphery (*Figure 2C–I'*, and *Figure 2—figure supplements 1 and 2*). These include: (1) impaired caudal migration of neurons comprising the VII[th] motor nucleus and a loss of motor innervation to the face (*Figure 2C–E'* and *Figure 2—figure supplement 2G*); (2) diminished staining of cholinergic nerves within the palate and retro-orbital fissure (*Figure 2—figure supplement 2D–F'*); and (3) decreased motor innervation of the tongue by the XII[th] nerve (*Figure 2F–G'*). The defect in migration of VII[th] motor neurons is consistent with that reported by *Qu et al. (2010)*. We also observed a nearly complete loss of cholinergic neurons in the vomeronasal organ (*Figure 2H–I'*). By contrast, the III[rd]-VI[th], X[th] and XII[th] motor nuclei appear unaltered in *ChAT-IRES-Cre*;*R26iAP*;*Fz3⁻/⁻* embryos and the trajectories of their axons appear to be normal (*Figure 2—figure supplements 1 and 2*).

The *ChAT-IRES-Cre*;*R26iAP* reporter showed decreased innervation of the *Fz3⁻/⁻* diaphragm (*Figure 2J–M'*), consistent with the analysis presented in *Figure 1D–H''*. The phrenic nerve defects may contribute to the respiratory distress that kills *Fz3⁻/⁻* neonates. A second contributing factor may be the disorganization of axons in the vicinity of the Pre-Bötzinger complex (data not shown), the brainstem region that coordinates respiratory movements by controlling the activity of the phrenic motor column (*Smith et al., 2009*).

## Impaired migration of a subset of neural crest cells in *Fz3⁻/⁻* embryos

By NF immunostaining of whole-mount E12 embryos, *Wang et al. (2006b)* observed a series of discrete NF-rich clusters along the caudal half of the *Fz3⁻/⁻* spinal cord that appear to represent ectopic neurons (*Figure 3A,A'*). Based on their dorsal midline location, we hypothesized that these clusters might arise from neural crest (NC) cells that had failed to migrate from their origin in the dorsal neural tube. To test this hypothesis, we asked whether cells in these clusters express transcription factors characteristic of NC derivatives. For example, Sox10 expression is detected in migrating NC cells and persists in melanocytes and a subset of dorsal root ganglion (DRG) neurons, and Brn3a and Islet1/2 are expressed in postmigratory DRG sensory neurons (*Marmigère and Ernfors, 2007*; *Sauka-Spengler and Bronner-Fraser, 2008*; *Bhatt et al., 2013*).

As shown in *Figure 3B–C''*, almost all of the cells in the NF-rich clusters express Sox10, Brn3a, or Islet1/2, consistent with a NC origin. Interestingly, Sox10 and Brn3a are expressed in mutually exclusive subsets of cells, as are Sox10 and Islet1/2, suggesting that despite their migratory defect, these presumptive NC cells are able to differentiate along appropriate committed lineages. Furthermore, Sox10 and NF immunolabeling within the dorsal clusters appear to be mutually exclusive or nearly so (*Figure 3E'',F'*), suggesting that at E11.5 the dorsal clusters consist of two classes of cells: immature NC cells at the migratory stage (Sox10⁺/NF⁻/Brn3a⁻/Islet1/2⁻) and more mature NC cells that are differentiating as DRG neurons (Sox10⁻/NF⁺/Brn3a⁺/Islet1/2⁺). These observations suggest that NC cell differentiation can proceed in the absence of cell migration, and that these dorsal clusters could serve as a useful model for dissecting extrinsic vs intrinsic influences on NC cell specification.

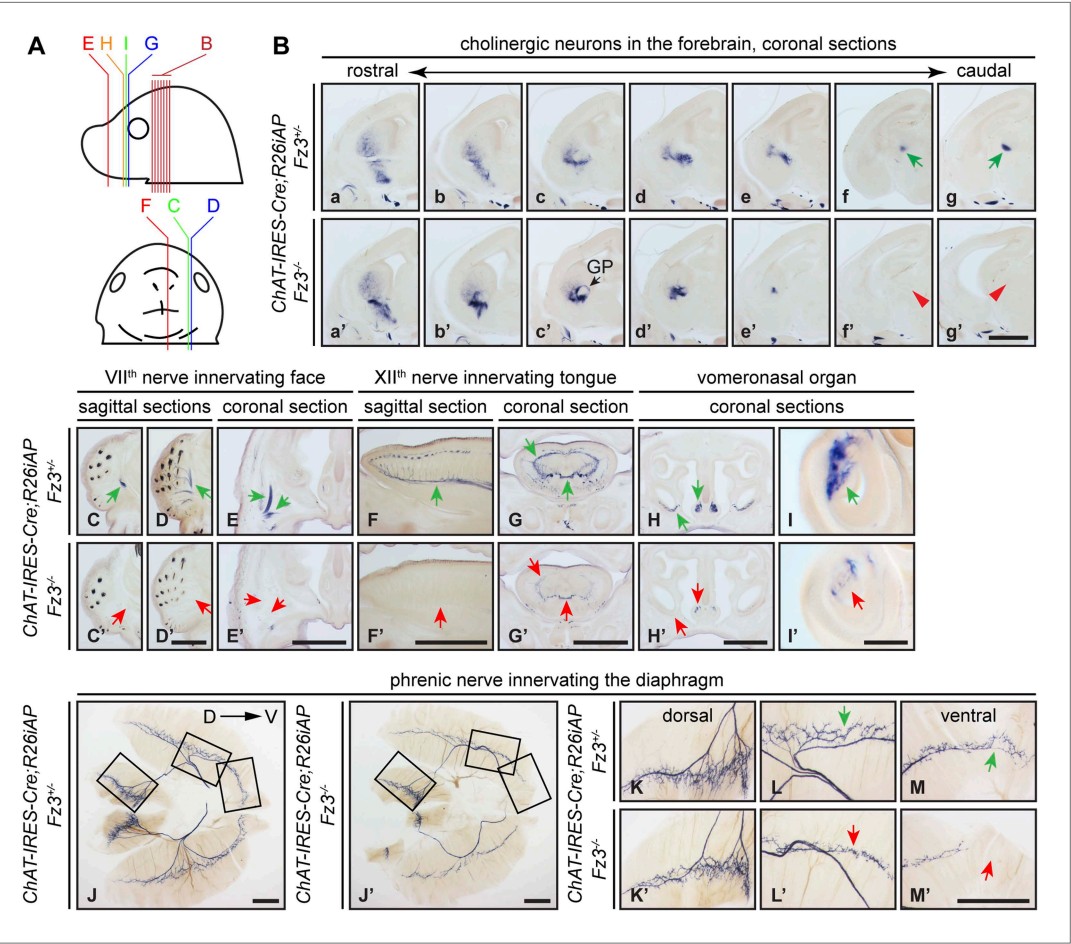

**Figure 2**. Diverse defects in cholinergic neurons shown by AP histochemisty in *ChAT-IRES-Cre;R26iAP;Fz3⁻/⁻* embryos. (**A**) Diagram showing the planes of sections from E18.5 heads in (**B–I'**). (**B**) Forebrain cholinergic neurons in consecutive coronal sections. The major cholinergic fiber tract passing through the striatum is missing in the *Fz3⁻/⁻* forebrain [green arrows in [**f**] and [**g**] vs red arrowheads in [**f'**] and [**g'**]]. GP, globus pallidus. Scale bar, 1 mm. (**C–E'**) Innervation of facial muscles by the VIIth nerve in sagittal (**C–D'**) and coronal (**E** and **E'**) sections. In the *Fz3⁻/⁻* head, facial muscles are not innervated (green arrows in [**C–E**] vs red arrows in [**C'–E'**]). Scale bars, 1 mm. (**F–G'**) Innervation of tongue musculature by the XIIth nerve in sagittal (**F** and **F'**) and coronal (**G** and **G'**) tongue sections. In *Fz3⁻/⁻* embryos, the number of axons is reduced (green arrows in [**F**] and [**G**] vs red arrows in [**F'**] and [**G'**]). Scale bars, 1 mm. (**H–I'**) Cholinergic neurons in the vomeronasal organ in coronal head sections. In *Fz3⁻/⁻* embryos, these neurons are markedly reduced (green arrows in [**H**] and [**I**] vs red arrows in [**H'**] and [**I'**]). Scale bars: (**H'**), 1 mm; (**I'**), 200 μm. (**J–M'**) Innervation of the diaphragm by the phrenic nerve visualized by AP histochemistry on flat-mount diaphragms. (**K–M**), enlarged views of boxed regions in (**J**); (**K'–M'**), enlarged views of boxed regions in (**J'**). Branching is diminished and number of motor terminals is reduced in the moderately affected *Fz3⁻/⁻* diaphragm (*Figure 1D–H*). D, dorsal; V, ventral. Scale bars, 1 mm.

The following figure supplements are available for figure 2:

**Figure supplement 1**. Oculomotor (IIIrd), trochlear (IVth), and abducens (VIth) nerves and their target innervation are not affected by loss of *Fz3*.

**Figure supplement 2**. The Vth (trigeminal) motor nerve and its target innervation are not affected by loss of *Fz3*.

We further observed that both the number and average size of the dorsal clusters decreased from E11.5 to E12.5 and they completely disappeared by ~E14.5, suggesting that the ectopic cells undergo programmed cell death. As shown in *Figure 3D–F''*, at E11.5 many cells in the dorsal clusters contain activated Caspase3, especially those that are Sox10⁻/NF⁺. Activated Caspase3 is largely absent from Sox10⁺/NF⁻ cells. Apparently the more differentiated DRG-like dorsal cluster neurons are the first to

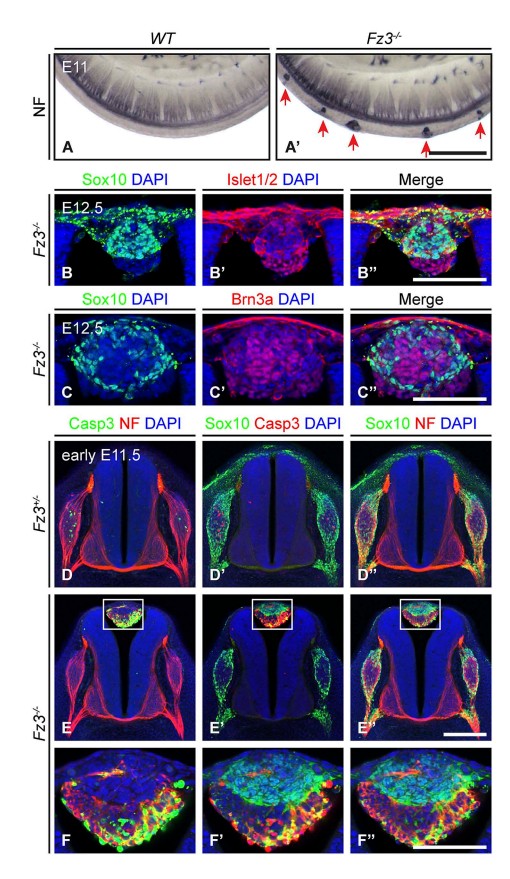

**Figure 3**. Neural crest cell migration defects in *Fz3*−/− mice. (**A** and **A'**) NF immunostaining of whole-mount E11 embryos showing aberrant clusters of neurons along the dorsal edge of the caudal neural tube in *Fz3*−/− embryos. Scale bar, 500 μm. (**B–C''**) Islet1/2, Brn3a, and Sox10 immunostaining in cross sections of the NF-rich clusters in *Fz3*−/− neural tubes at E12.5. Scale bars, 100 μm. (**D–F''**) NF, cleaved Caspase3, and Sox10 immunostaining in cross sections of early E11.5 *Fz3*+/− (**D–D''**) and *Fz3*−/− (**E–E''**) spinal cords. The boxed regions in (**E–E''**) are enlarged in (**F–F''**). Scale bars: (**E''**), 200 μm; (**F''**), 100 μm.

undergo apoptotic cell death, and the less differentiated NC derivatives are temporarily spared that fate.

In the lumbar region of E11.5 *Fz3*−/− embryos, there is a ~0.5 day delay in the appearance of apoptotic neurons within the DRG, as determined by immunostaining for activated Caspase3, relative to *Fz3*+/− littermates (compare *Figure 3D,D'* vs *Figure 3E,E'*). This delay in apoptotic death among DRG neurons is not observed in the cervical region of E11.5 *Fz3*−/− embryos, which is devoid of dorsal clusters. In DRGs, NC cells differentiate into neurons in greater numbers than are required for full target innervation, and the resulting competition for target-derived survival factors leads to the death of excess neurons (*Oppenheim, 1991*; *Cowan, 2001*). Since apoptotic cell death in *WT* DRGs is well underway by E11.5, when most DRG axons are still far from their final targets, it is likely that intermediate targets are a major source of survival signals. The similar distributions along the spinal cord of dorsal clusters and delayed apoptosis suggested the possibility that in the lumber region of *Fz3*−/− embryos there may be an initial reduction in the number of NC cells arriving at the DRG, leading to decreased competition for peripheral targets and higher rates of survival of DRG neurons.

## *Fz3* loss-of-function results in severe thinning of the spinal motor nerve innervating the dorsal limb

In light of the essential role of *Fz3* in the development of subsets of cranial motor neurons, we evaluated the role of *Fz3* in the development of spinal motor neurons that innervate the limbs, a set for which the axonal projection patterns have been well defined (*De Marco Garcia and Jessell, 2008*; *Bonanomi and Pfaff, 2010*). NF immunostaining of E11.5-E13.5 limbs showed that the nerve innervating the dorsal limb is moderately thinned in *Fz3*−/− forelimbs and moderately to severely thinned in *Fz3*−/− hindlimbs (*Figure 4A–H'*, and *Figure 4—figure supplements 1 and 2*). In the hindlimb, roughly 50% of *Fz3*−/− embryos exhibit a moderate defect, with a thinned dorsal nerve that is still visible along its distal trajectory. The remaining ~50% of *Fz3*−/− embryos exhibit a severe defect, with a near absence of the dorsal nerve. Other nerves within the limb appear to be largely unaffected.

To ascertain the motor vs sensory identity of the affected axons, we crossed the *Hb9-EGFP* transgene to *Fz3* mutant mice to selectively label motor axons. GFP immunostaining of *Hb9-EGFP;Fz3*+/+ and *Hb9-EGFP;Fz3*−/− limbs at E12.5 showed that motor axons are severely under-represented in the dorsal nerve in both fore- and hindlimbs in mutant embryos (*Figure 4I–L''*). The affected axons derive from motor neurons in the lateral division of the lateral motor column (LMC_L; *Figure 4A*).

Close examination of the pattern of NF immunostaining showed that the *Fz3*−/− defect occurs distal to the point where LMC axons separate into dorsal and ventral nerves. More specifically, in *Fz3*−/− limbs an abrupt decrease in diameter is observed in the dorsal nerve immediately distal to the emergence of its first major side branch (*Figure 4B'–H',M*). Therefore, the defect in the *Fz3*−/− limb appears to be

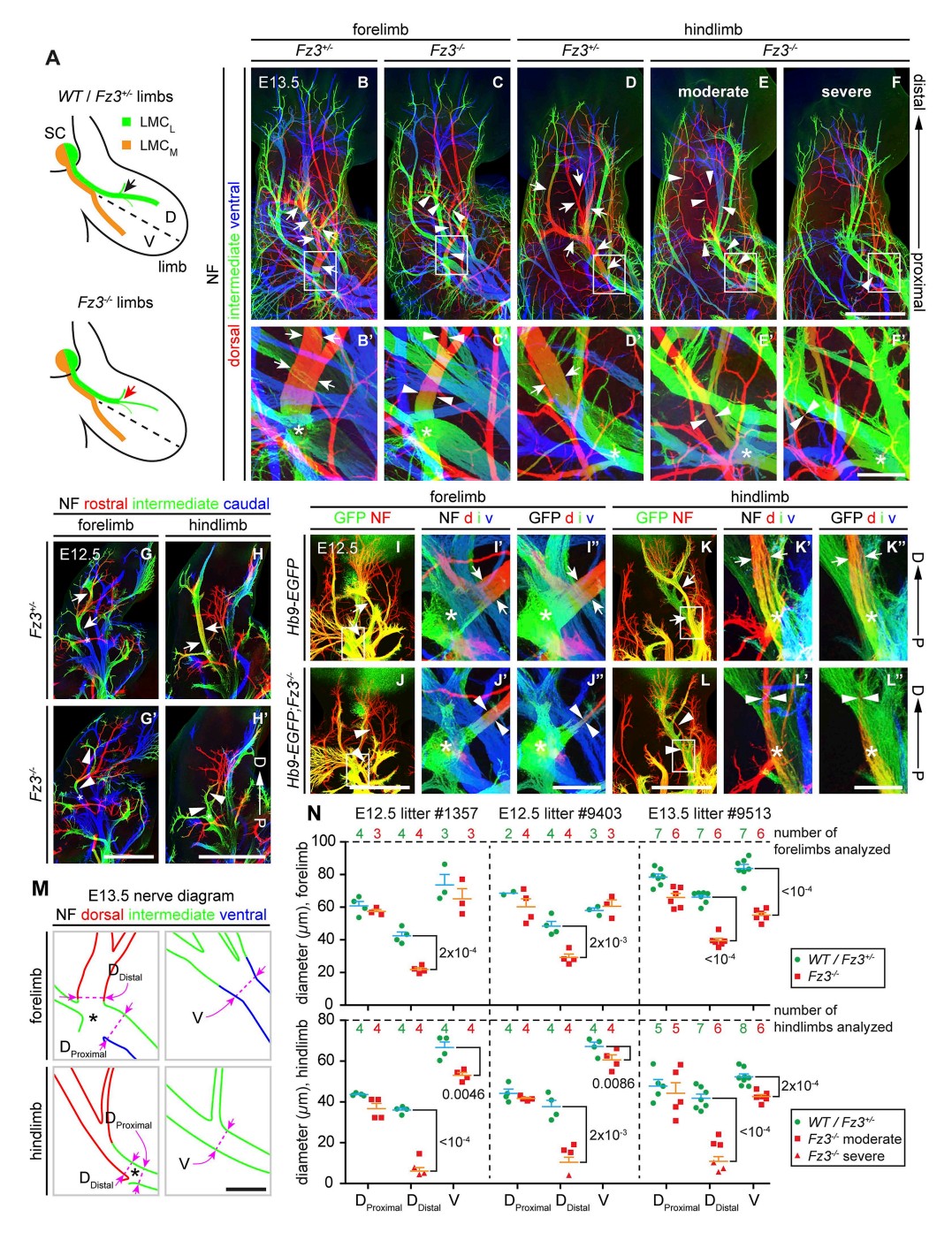

**Figure 4**. Thinning of the dorsal motor nerve in *Fz3⁻/⁻* limbs. (**A**) Diagram showing dorsal (D) and ventral (V) limb innervation by spinal motor nerves in control (*WT* or *Fz3⁺/⁻*; top) and *Fz3⁻/⁻* (bottom) embryos. In *Fz3⁻/⁻* limbs, the dorsal nerve is thinned distal to the plexus at the base of the limb (highlighted by arrows). SC, spinal cord. (**B–F'**) NF immunostaining of whole-mount fore- and hindlimbs from E13.5 *Fz3⁺/⁻* and *Fz3⁻/⁻* embryos. (**B'–F'**), magnified view of boxed regions in (**B–F**). Arrows (control) and arrowheads (mutant) indicate dorsal nerves, which are moderately thinner in *Fz3⁻/⁻* forelimbs and substantially thinner in *Fz3⁻/⁻* hindlimbs. Asterisks indicate the points at which axons stall in *Fz3⁻/⁻* limbs. The red to blue color code represents depth within the Z-stack, oriented here along the dorsal to ventral axis. Scale bars: (**B–F**), 500 *μm*; (**B'–F'**), 100 *μm*. (**G–H'**) NF immunostaining of 1 mm-thick sections in the dorsoventral plane encompassing fore- or hindlimbs from E12.5 *Fz3⁺/⁻* and *Fz3⁻/⁻* embryos. Arrows (control) and arrowheads (*Fz3⁻/⁻*) indicate dorsal nerves, which are thinner in *Fz3⁻/⁻* limbs. Asterisks indicate the
*Figure 4. Continued on next page*

*Figure 4. Continued*

points at which axons stall in *Fz3*$^{-/-}$ limbs. Red to blue color code represents depth within the Z-stack, oriented here along the rostrocaudal axis. P, proximal; D, distal. Scale bars, 500 *μm*. (**I**–**L''**) GFP (green) and NF (red) double immunostaining of whole-mount forelimbs and hindlimbs from E12.5 *Hb9-EGFP;Fz3*$^{+/+}$ and *Hb9-EGFP;Fz3*$^{-/-}$ embryos (**I**–**L**). Motor axons (labeled with GFP) are selectively eliminated from the dorsal nerve in mutant limbs (arrowheads and arrows). (**I'**–**L'**) are enlarged views of the NF immunostaining in the boxed regions in (**I**–**L**), with color representing depth along the dorsoventral axis. Similarly, (**I''**–**L''**) are enlarged views of the GFP immunostaining. P, proximal; D, distal; d, dorsal; i, intermediate; v, ventral. Scale bars: (**J**) and (**L**), 500 *μm*; (**J''**) and (**L''**), 100 *μm*. (**M**) Diagrams showing the locations of nerve diameter measurements quantified in (**N**), traced from images (**B'**) and (**D'**). The locations of two dorsal nerve measurements (D$_{Distal}$, dorsal nerve diameter measured immediately distal to the stalling point; D$_{Proximal}$, dorsal nerve diameter measured immediately proximal to the stalling point) and one ventral nerve measurement (V) are shown. Scale bar, 100 *μm*. (**N**) The diameters of motor nerves at the locations shown in (**M**) were measured from NF immunostained whole-mount forelimbs (top) and hindlimbs (bottom) from two litters at E12.5 and one litter at E13.5. Bars, mean ± SEM. p-values are shown for the indicated pair-wise comparisons (student's *t*-test).

The following figure supplements are available for figure 4:

**Figure supplement 1**. Thinning of spinal motor nerves innervating the dorsal forelimb.

**Figure supplement 2**. Thinning of spinal motor nerves innervating the dorsal hindlimb.

**Figure supplement 3**. Loss of *Fz3* does not affect axial or body wall motor nerve trajectories.

---

a stalling of LMC$_L$ axons at this location. To quantify this phenotype, we measured the thickness of the dorsal nerve proximal and distal to this point. We also measured the thickness of the ventral nerve at the same distance from the plexus (*Figure 4M*). All measurements were made on NF-immunostained whole-mount limbs viewed in a dorsal-to-ventral orientation. *Figure 4N* shows the quantification of nerve thickness for fore- and hindlimbs among littermates in two E12.5 litters and one E13.5 litter. In *Figure 4N*, moderate and severe phenotypes within the hindlimb are represented by distinct symbols, but the statistical analysis ignores this distinction and simply compares *Fz3*$^{+/-}$ and *Fz3*$^{+/+}$ vs *Fz3*$^{-/-}$ littermates. For both the fore- and hindlimbs in *Fz3*$^{-/-}$ embryos, several findings emerge: (1) the thinning of the dorsal nerve is highly significant statistically and it occurs over a remarkably small interval, between the regions labeled D$_{Proximal}$ and D$_{Distal}$ in *Figure 4M*; (2) proximal to position D$_{Proximal}$, dorsal nerve thinning is minimal and statistically insignificant; (3) the ventral nerve exhibits a small decrease in thickness in 5/6 comparisons, with 4/6 showing a difference that is statistically significant (p<0.05), implying that dorsal nerve axons have not rerouted to the ventral nerve, which would have then shown an increase in thickness; and (4) the average nerve thickness, relative to the littermate controls, is 31% for the dorsal nerve in the hindlimb, 57% for the dorsal nerve in the forelimb, and 82% for the ventral nerve when averaged over fore- and hindlimbs.

In *Fz3*$^{-/-}$ embryos, other classes of spinal motor axons appear to be unaffected, including medial motor column and hypaxial motor column axons that innervate axial muscles and body wall muscles, respectively (*Figure 4—figure supplement 3*). Although *Fz3* is widely expressed in the mouse spinal cord and brainstem, it appears to be uniquely required by only a subset of motor neurons.

## *Fz3* controls axon growth of LMC$_L$ motor neurons

We next asked whether the primary cause of motor axon deficiency in the VII[th], XII[th], phrenic, and dorsal limb nerves in *Fz3*$^{-/-}$ mice is: (1) defective motor neuron differentiation, (2) excessive motor neuron death, or (3) defective axon growth and/or guidance. Since the vast majority of *Fz3*$^{-/-}$ motor axons do not appear to follow erroneous trajectories (*Figures 1A,A', and 4*), aberrant path-finding is unlikely to be relevant. There are also no apparent defects in motor axon fasciculation in *Fz3*$^{-/-}$ embryos (*Figures 1 and 4*). To distinguish among the remaining possibilities, we focused on LMC$_L$ motor neurons and their axons, because (1) these neurons can be unambiguously identified based on their patterns of transcription factor expression; (2) their axons can be clearly identified within the limb; and (3) several signaling pathways that control LMC$_L$ axon growth and guidance decisions provide points of comparison.

In the developing mouse spinal cord, postmitotic motor neurons are generated between E9 and E11 (*Arber et al., 1999*). Motor neurons are divided into pools positioned at stereotypical locations

along the rostrocaudal, dorsoventral, and mediolateral axes of the spinal cord. Each motor pool innervates specific peripheral targets and can be characterized by the expression of various transcription factors (**Bonanomi and Pfaff, 2010**). For example, Islet1 is expressed by $LMC_M$ motor neurons and Lhx1 is expressed by $LMC_L$ motor neurons, while Islet2 and Foxp1 are pan-LMC markers (**Figure 5A**). By using Islet1 and Foxp1 double immunostaining on cross sections from E11.5 lumbar-level spinal cords we observed that the $LMC_L$ pool (Foxp1$^+$/Islet1$^-$) is present in $Fz3^{-/-}$ mice (**Figure 5B,B'**). This result implies that *Fz3* is not required for the initial specification of $LMC_L$ motor neurons.

To quantitatively assess motor neuron populations at a time when the $Fz3^{-/-}$ phenotype is fully manifest, we counted the number of $LMC_L$ neurons in serial cross sections from E12.5 lumbar-level spinal cords following Islet1 and Foxp1 immunostaining of three pairs of control and $Fz3^{-/-}$ littermate embryos. As both $LMC_M$ and $LMC_L$ neurons were counted in the same sections, the number of $LMC_M$ neurons served as an internal control. As shown in **Figure 5C,C'**, in the region of the $Fz3^{-/-}$ spinal cord that contains $LMC_L$ neurons, the number of $LMC_L$ neurons was significantly decreased, while the number of $LMC_M$ neurons was similar to that in $Fz3^{+/-}$ controls (sections 1–6). In contrast, the numbers of both $LMC_M$ and $LMC_L$ neurons in more rostral regions were unaffected by loss of *Fz3* (sections 7–12).

The decrease in $LMC_L$ neurons at E12.5 suggests that loss of *Fz3* could lead—directly or indirectly—to $LMC_L$ motor neuron death. To test this possibility, we performed cleaved Caspase3 immunostaining on lumbar spinal cord sections from embryos at E11.5 and E12.5, developmental stages prior to the wave of naturally occurring motor neuron death at E13.5-E15.5 (**Kablar and Rudnicki, 1999**; **Oppenheim et al., 2000**). As expected, very few apoptotic cells were observed in $Fz3^{+/-}$ controls at E11.5 and E12.5. In contrast, massive cell death is seen in the region corresponding to the LMC in the brachial and lumbar regions of the $Fz3^{-/-}$ spinal cord (**Figure 5D–E''** and **Figure 5—figure supplement 1A–D''**). By E12.5 the lumbar LMC volume is substantially smaller in $Fz3^{-/-}$ compared with $Fz3^{+/-}$ spinal cords (**Figure 5F–G''**). Similarly, in the VII$^{th}$ and XII$^{th}$ cranial nerve nuclei at E12.5, there is an analogous wave of precocious motor neuron death that is not observed in $Fz3^{+/-}$ controls (**Figure 5H–K'**). Thus, in $Fz3^{-/-}$ embryos a discrete subset of motor neurons exhibits both axon growth defects and apoptotic cell death at E11.5-E12.5.

The results thus far indicate that *Fz3* inactivation could lead either to a primary defect in motor neuron viability that secondarily impairs axonal growth or a primary defect in axonal growth that secondarily impairs motor neuron viability. To distinguish these two possibilities, we used a $Bax^{-/-}$ background—in which loss of the proapoptotic molecule *Bax* largely eliminates apoptosis of motor neurons (**White et al., 1998**)—to ask whether blocking (or delaying) motor neuron apoptosis rescues the $Fz3^{-/-}$ dorsal nerve defect. Although the precocious motor neuron death seen in $Fz3^{-/-}$ embryos was eliminated in $Bax^{-/-};Fz3^{-/-}$ embryos, the dorsal nerve defect at the base of the limbs remained unchanged (**Figure 5M–R'** and **Figure 5—figure supplement 1E–G'''**). Taken together, these data imply that in spinal motor neurons the primary phenotype resulting from loss of *Fz3* is a defect in the growth of $LMC_L$ axons. Since apoptosis is restricted to precisely those motor neurons that exhibit defective axonal growth and since the axonal growth defect is not secondary to apoptosis of $LMC_L$ neurons, it seems very likely that the chain of causality leads from the axonal growth defect to apoptosis.

## *Fz3* expression is required in the CNS but not in the periphery to control axon growth

We next asked whether the axonal phenotypes in $Fz3^{-/-}$ embryos reflect *Fz3* action exclusively in CNS neurons. Based on the pattern of *lacZ* reporter expression from the original *Fz3* null allele (which carries an in-frame *lacZ* open reading frame), *Fz3* expression appears to be far higher in the CNS than in other tissues (**Wang et al., 2002**). However, rigorously answering this question requires selective inactivation of *Fz3* in defined cell populations. To that end, we generated a *Fz3* conditional allele ($Fz3^{CKO}$), in which *LoxP* sites flank exon 3, the largest coding exon, within the *Fz3* gene (**Figure 6—figure supplement 1A,B**). Following Cre-mediated recombination, the exon 3-deleted allele behaves as a null (data not shown). To eliminate *Fz3* expression exclusively in CNS neurons, we controlled recombination with an $Olig2^{Cre}$ knock-in allele and found that $Olig2^{Cre/+};Fz3^{CKO/-}$ mice exhibit the complete spectrum of motor neuron phenotypes shown by $Fz3^{-/-}$ mice: (1) VII$^{th}$ nerve axons do not form the well-defined C-shaped tract looping around the VI$^{th}$ cranial nerve nucleus (**Figure 6A,A'**); (2) the XII$^{th}$ cranial nerve is markedly thinned and fails to innervate the tongue at E13.5 (**Figure 6B–C'**); (3) the phrenic nerve is thinned (**Figure 6D,D'**); and (4) the dorsal nerves innervating the fore- and hindlimbs are thinned (**Figure 6E–H'**). This experiment rules out any role for Fz3 in peripheral tissues. *Fz3* expression

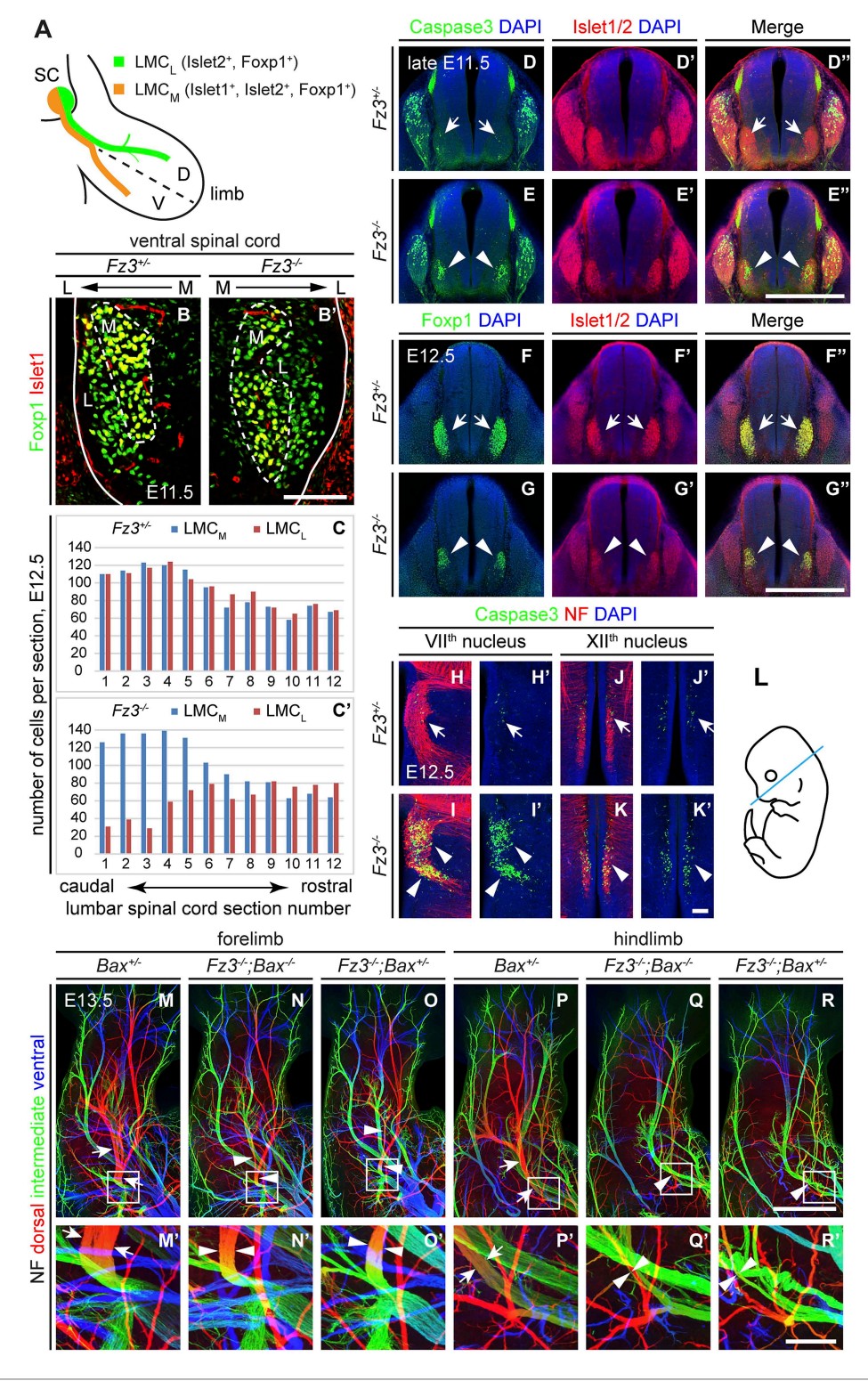

**Figure 5**. Abundance, differentiation, and apoptosis of motor neurons in *Fz3⁻/⁻* embryos, and the effects of suppressing apoptosis. (**A**) Diagram showing the pattern of transcription factor expression in LMC$_L$ and LMC$_M$ motor neurons. SC, spinal cord; D, dorsal; V, ventral. (**B** and **B'**) Islet1 and Foxp1 expression in spinal motor neurons in cross sections of E11.5 *Fz3⁺/⁻* and *Fz3⁻/⁻* lumbar spinal cords. Continuous white line, lateral edge of the spinal cord. The broken white line encircles LMC$_M$ motor neurons. L, lateral; M, medial. Scale bar, 100 µm. (**C** and **C'**) The
*Figure 5. Continued on next page*

*Figure 5. Continued*

number of LMC_L (Islet1⁻/Foxp1⁺) and LMC_M (Islet1⁺/Foxp1⁺) motor neurons per 14 µm frozen section in E12.5 *Fz3⁺/⁻* and *Fz3⁻/⁻* lumbar spinal cords. Motor neurons were counted and averaged from 12 serial sections from each of three pairs of embryos, with adjacent counted sections separated by four uncounted sections. (**D–E″**) Motor neuron apoptosis visualized with Islet1/2 and cleaved Caspase3 immunostaining in cross sections of E11.5 *Fz3⁺/⁻* (arrows) and *Fz3⁻/⁻* (arrowheads) lumbar spinal cords. Scale bar, 500 µm. (**F–G″**) Reduced LMC volume in the *Fz3⁻/⁻* lumbar spinal cord visualized by comparing Islet1/2 and Foxp1 immunostaining in cross sections of E12.5 *Fz3⁺/⁻* (arrows) and *Fz3⁻/⁻* (arrowheads) lumbar spinal cords. Scale bar, 500 µm. (**H–L**) Cell death in VII^th (**H–I′**), and XII^th (**J–K′**) cranial motor nuclei (arrows and arrowheads) visualized with cleaved Caspase3 and NF immunostaining in horizontal sections of E12.5 *Fz3⁺/⁻* and *Fz3⁻/⁻* embryos. (**H′–K′**) show cleaved Caspase3 immunostaining with DAPI counterstaining for (**H–K**). (**L**) planes of section. Scale bar, 100 µm. (**M–R′**) NF immunostaining of whole-mount forelimbs (**M–O′**) and hindlimbs (**P–R′**) from E12.5 *Bax⁺/⁻*, *Fz3⁻/⁻;Bax⁻/⁻*, and *Fz3⁻/⁻;Bax⁺/⁻* embryos. For each pair of panels, the inset in the upper panel is enlarged in the lower panel. Arrows and arrowheads indicate the dorsal nerve. Depth is color coded as in ***Figure 4B–F′***. Scale bars: (**R**), 500 µm; (**R′**), 100 µm.

The following figure supplements are available for figure 5:

**Figure supplement 1**. Precocious motor neuron death in *Fz3⁻/⁻* spinal cord is blocked by loss of *Bax*.

in DRG neurons also appears to be irrelevant since *Olig2^Cre* is not expressed in DRG neurons (***Huettl et al., 2011***; ***Chen and Wichterle, 2012***), a conclusion that was further confirmed by inactivating the *Fz3^CKO* allele specifically in DRG neurons with a *Wnt1-Cre* transgene (*Wnt1-Cre;Fz3^CKO/⁻*; ***Danielian et al., 1998***) and observing that the dorsal nerves innervating the fore- and hindlimbs were unaffected (***Figure 6—figure supplement 1C–D′***). These data imply that the LMC_L motor axon defect is not an indirect consequence of Fz3-dependent effects on co-fasciculating sensory axons.

## Defects in peroneal nerve development in *Fz3⁻/⁻* mice lead to atrophy of anterior compartment muscles in the distal hindlimb

The perinatal death of *Fz3⁻/⁻* mice has, until now, prevented an analysis of the *Fz3⁻/⁻* phenotype in postnatal life. To circumvent this problem, we combined a *Caudal-1 (Cdx1)-Cre* transgene with the *Fz3^CKO* allele to inactivate *Fz3* in all tissues caudal to the upper thorax before midgestation (***Hierholzer and Kemler, 2009***). The postnatal viability of *Cdx1-Cre;Fz3^CKO/⁻* mice implies that structures critical for breathing and feeding develop normally. Like *Fz3⁻/⁻* embryos at E18.5, *Cdx1-Cre;Fz3^CKO/⁻* mice exhibit plantar-flexed hindlimbs and a curled tail (***Figure 7A***; compare to ***Figure 2*** of ***Wang et al., 2002***). In *Cdx1-Cre;Fz3^CKO/⁻* embryos at E12.5, the thinning of the dorsal nerve in fore- and hindlimbs is indistinguishable from the thinning observed in *Fz3⁻/⁻* embryos (***Figure 7B–E′***).

*Cdx1-Cre;Fz3^CKO/⁻* mice ambulate primarily with their forelimbs, which appear to be grossly normal, suggesting that the moderate thinning of the dorsal nerve in the forelimb does not seriously compromise innervation of the musculature. In cross sections of distal hind legs in *Hb9-EGFP;Fz3⁻/⁻* embryos at E15.5, we observed a selective loss of motor innervation of the muscles of the anterior compartment (tibialis anterior, extensor hallucis longus, and extensor digitorum longus muscles) by the deep peroneal nerve (***Figure 7F,F′***), and in adult *Cdx1-Cre;Fz3^CKO/⁻* mice this same group of muscles is selectively and completely atrophied as determined by magnetic resonance imaging (***Figure 7G,G′***).

The severe plantar flexion of *Fz3⁻/⁻* and *Cdx1-Cre;Fz3^CKO/⁻* hindfeet is readily understood from the anatomy of the limb musculature and its innervation (***Gray and Lewis, 1918***). The dorsal muscles in the embryonic limb become extensors and abductors, which are innervated by the peroneal nerve, while the ventral muscles become flexors and adductors, which are innervated by the tibial nerve. After the developing hindlimb undergoes a 90-degree rotation—which rotates the plantar surface of the foot away from the midline—the dorsal limb muscles face anteriorly and ventral limb muscles face posteriorly. Failure of dorsal nerve development affects muscles in the anterior compartment, which dorsiflex the foot. As a result, the normal balance between antagonistic flexor and extensor muscle groups is lost, and the action of the unopposed extensors leads to extreme plantar flexion of the foot.

## Comparison of innervation defects in *Fz3⁻/⁻* and *Ret⁻/⁻* limbs

As noted in the Introduction, defects in motor axon guidance in the dorsal limb have been described in *EphA4⁻/⁻*, *Gdnf⁻/⁻*, and *Ret⁻/⁻* embryos, and in each case dorsal motor axons are misrouted to the ventral limb. To compare one example of this class of mutants side-by-side with *Fz3⁻/⁻* and to examine

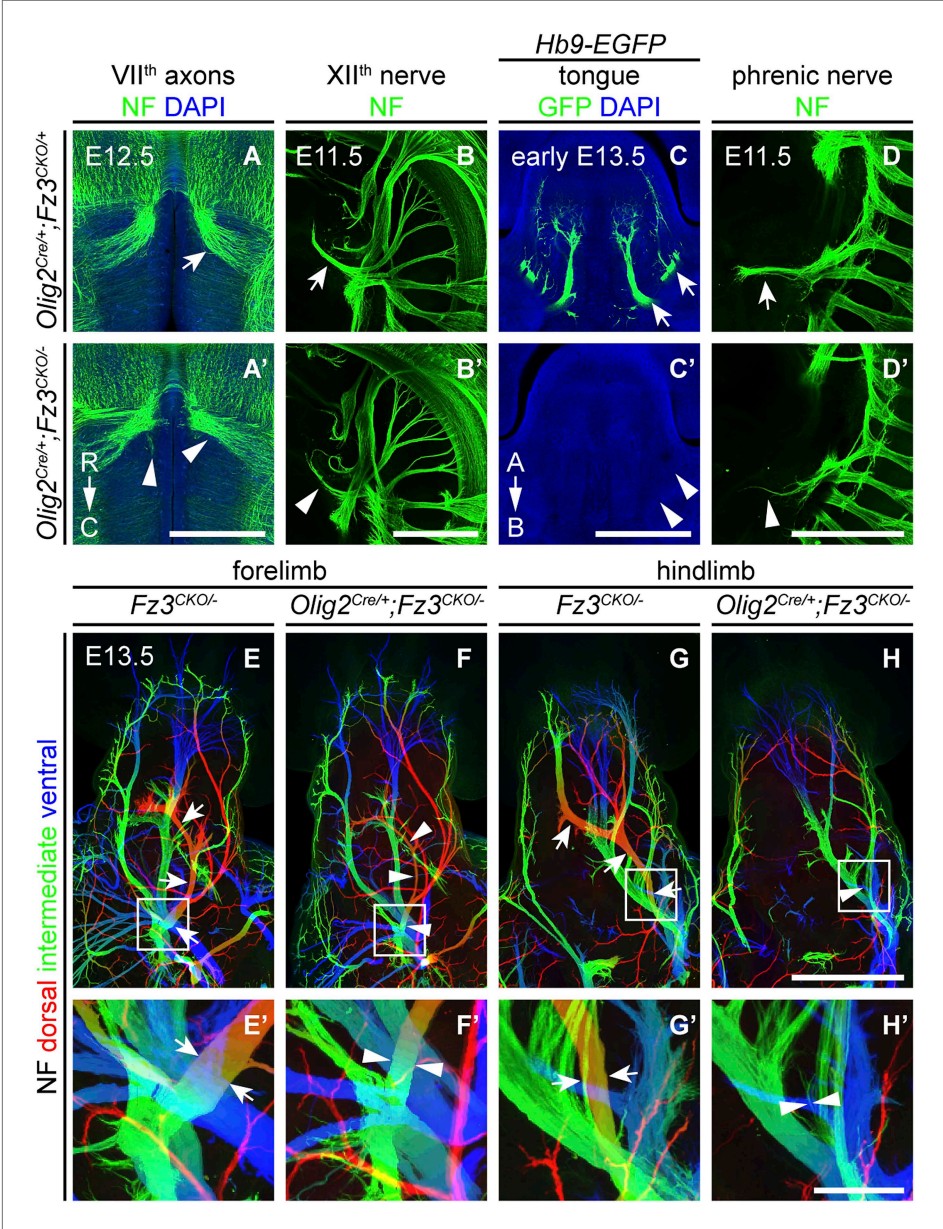

**Figure 6**. *Olig2^{Cre/+};Fz3^{CKO/−}* embryos recapitulate the motor neuron phenotype of *Fz3^{−/−}* embryos. (**A** and **A'**) The VII^{th} cranial nerve axons visualized by NF immunostaining of horizontal sections through E12.5 *Olig2^{Cre/+};Fz3^{CKO/+}* (arrow) and *Olig2^{Cre/+};Fz3^{CKO/−}* (arrowhead) brainstems. In the *Olig2^{Cre/+};Fz3^{CKO/−}* brainstem, the more rostromedial locations of the VII^{th} nerve cell bodies are apparent, and the VII^{th} nerve axons fail to loop around the VI^{th} cranial nerve nucleus. C, caudal; R, rostral. Scale bar, 500 *μm*. (**B** and **B'**) The XII^{th} cranial nerve visualized by NF immunostaining in lateral views of whole-mount E11.5 *Olig2^{Cre/+};Fz3^{CKO/+}* (arrow) and *Olig2^{Cre/+};Fz3^{CKO/−}* (arrowhead) embryos. The XII^{th} nerve has a reduced diameter in *Olig2^{Cre/+};Fz3^{CKO/−}* embryos. Scale bar, 500 *μm*. (**C** and **C'**) Innervation of tongue muscles by the XII^{th} cranial nerve visualized by GFP immunostaining in coronal sections of early E13.5 *Hb9-EGFP;Olig2^{Cre/+};Fz3^{CKO/+}* (arrows) and *Hb9-EGFP;Olig2^{Cre/+};Fz3^{CKO/−}* (arrowheads) embryos. The *Hb9-EGFP;Olig2^{Cre/+};Fz3^{CKO/−}* tongue shows no motor innervation. A, apical; B, basal. Scale bar, 500 *μm*. (**D** and **D'**) The phrenic nerve visualized by NF immunostaining in lateral views of whole-mount E11.5 *Olig2^{Cre/+};Fz3^{CKO/+}* (arrow) and *Olig2^{Cre/+};Fz3^{CKO/−}* (arrowhead) embryos. The phrenic nerve has a reduced diameter in *Olig2^{Cre/+};Fz3^{CKO/−}* embryos. Scale bar, 500 *μm*. (**E–H'**) NF immunostaining of whole-mount forelimbs (**E** and **F**) and hindlimbs (**G** and **H**) from E12.5 *Fz3^{CKO/−}* and *Olig2^{Cre/+};Fz3^{CKO/−}* embryos. Boxed regions in (**E–H**) are enlarged in (**E'–H'**). Arrows and arrowheads indicate the dorsal nerve. Depth is color coded as in *Figure 4B–F*. Scale bars: (**H**), 500 *μm*; (**H'**), 100 *μm*.

*Figure 6. Continued on next page*

*Figure 6. Continued*

The following figure supplements are available for figure 6:

**Figure supplement 1**. Design of the *Fz3*^CKO allele, and limb innervation in *Wnt1-Cre;Fz3*^CKO/− mice.

---

the possibility of a genetic interaction between the Fz3 and the GDNF/Ret signaling pathways, we examined limb innervation phenotypes in *Ret*^−/−, *Fz3*^+/−;*Ret*^+/−, and *Fz3*^−/−;*Ret*^+/− embryos (*Figure 7H–J'*). This analysis provides no evidence for a genetic interaction between *Fz3* and *Ret*, as *Fz3*^+/−;*Ret*^+/− hindlimb innervation is indistinguishable from *WT*, and *Fz3*^−/−;*Ret*^+/− hindlimb innervation is indistinguishable from *Fz3*^−/− (compare *Figures 4B–F' and 7H–J'*, *Figure 4—figure supplements 1 and 2*, and *Figure 7—figure supplement 1A–C'*). Our analysis of *Ret*^−/− fore- and hindlimb innervation fully confirms previous descriptions (*Kramer et al., 2006*) and shows that in *Ret*^−/− hindlimbs the dorsal nerve is thinned to a similar extent both proximal and distal to the point at which *Fz3*^−/− dorsal motor axons stall (asterisks in *Figure 7H'–J'*) and the ventral nerve is thicker than its counterpart in *WT* littermates, consistent with a rerouting of dorsal motor axons to the ventral nerve. These experiments confirm that the *Fz3*^−/− dorsal motor axon phenotype is anatomically distinct from the phenotypes described in *EphA4*^−/−, *Gdnf*^−/−, and *Ret*^−/− limbs (*Figure 7K*).

## Discussion

The experiments reported here establish a role for Frizzled signaling in the growth of a subset of cranial and spinal motor axons. In the absence of Fz3, innervation is decreased or entirely missing from the musculature of the tongue, snout, diaphragm, and the dorsal (which will later become the anterior) compartment of the lower hindlimbs. In the limb, the *Fz3*^−/− phenotype is distinct from several previously described axon guidance phenotypes. *Fz3*^−/− LMC_L axons, which are normally destined to innervate the dorsal limb musculature, stall abruptly within the plexus and do not adopt a default trajectory as observed in embryos with loss of EphrinA/EphA4, Sema3F/Npn2, and GDNF/Ret signaling. The loss of Fz3 also leads to the precocious apoptotic death of affected cranial and spinal motor neurons, a phenotype not observed in embryos with defective EphrinA/EphA4 and GDNF/Ret signaling (*Helmbacher et al., 2000*; *Kramer et al., 2006*). To the best of our knowledge, the only previously reported mammalian axon guidance phenotype that might resemble the *Fz3*^−/− phenotype to some extent is that seen in the absence of *Linx*, a transmembrane protein of the LIG family that has been implicated as a cofactor in receptor tyrosine kinase signaling (*MacLaren et al., 2004*; *Mandai et al., 2009*).

These observations raise a number of fascinating questions. Since Fz3 is widely expressed in the developing CNS, why are the motor neuron phenotypes restricted to a subset of motor axons? Are other Frizzled receptors playing analogous or redundant roles in other motor axons? Do signaling molecules produced by target tissues—for example, Wnts—relay positional information to the axon via PCP proteins? Is there an interaction between PCP signaling and other axon guidance systems? By what mechanism is information from PCP proteins at the plasma membrane communicated to the axon's cytoskeleton to control axonal growth? What is the survival signal that keeps *WT* LMC_L motor neurons alive at E11-E13, but fails to act on *Fz3*^−/− LMC_L motor neurons?

### Cre lines that permit postnatal survival by localizing nervous system defects to caudal territories

Of technical interest is our use of Cre-mediated recombination in the caudal ~80% of the embryo as a strategy for studying genes that are required more rostrally for survival (e.g., genes required for brain development). The *Cdx1-Cre* transgene used here recombines *LoxP* targets at ~100% efficiency by ~E8 in all solid tissues caudal to the upper thorax (*Hierholzer and Kemler, 2009*). A *Cdx2-Cre* transgene produces a similar pattern of recombination except that the most rostral extent of recombination is located roughly at the umbilicus (*Hinoi et al., 2007*). Thus, mice harboring a conditional allele that impairs cardiac or pulmonary function would likely be spared if the allele was recombined with the *Cdx2-Cre* transgene. These two *Cre* lines should be generally useful for studying spinal cord, DRG, and peripheral nerve phenotypes in postnatal mice harboring conditional alleles in a wide variety of genes.

### Fz3 and neural crest migration

NC cell migration is one of the most dramatic migration processes in vertebrate development. At mid-gestation, NC cells delaminate from the dorsal neuro-ectoderm to populate skin (melanocytes),

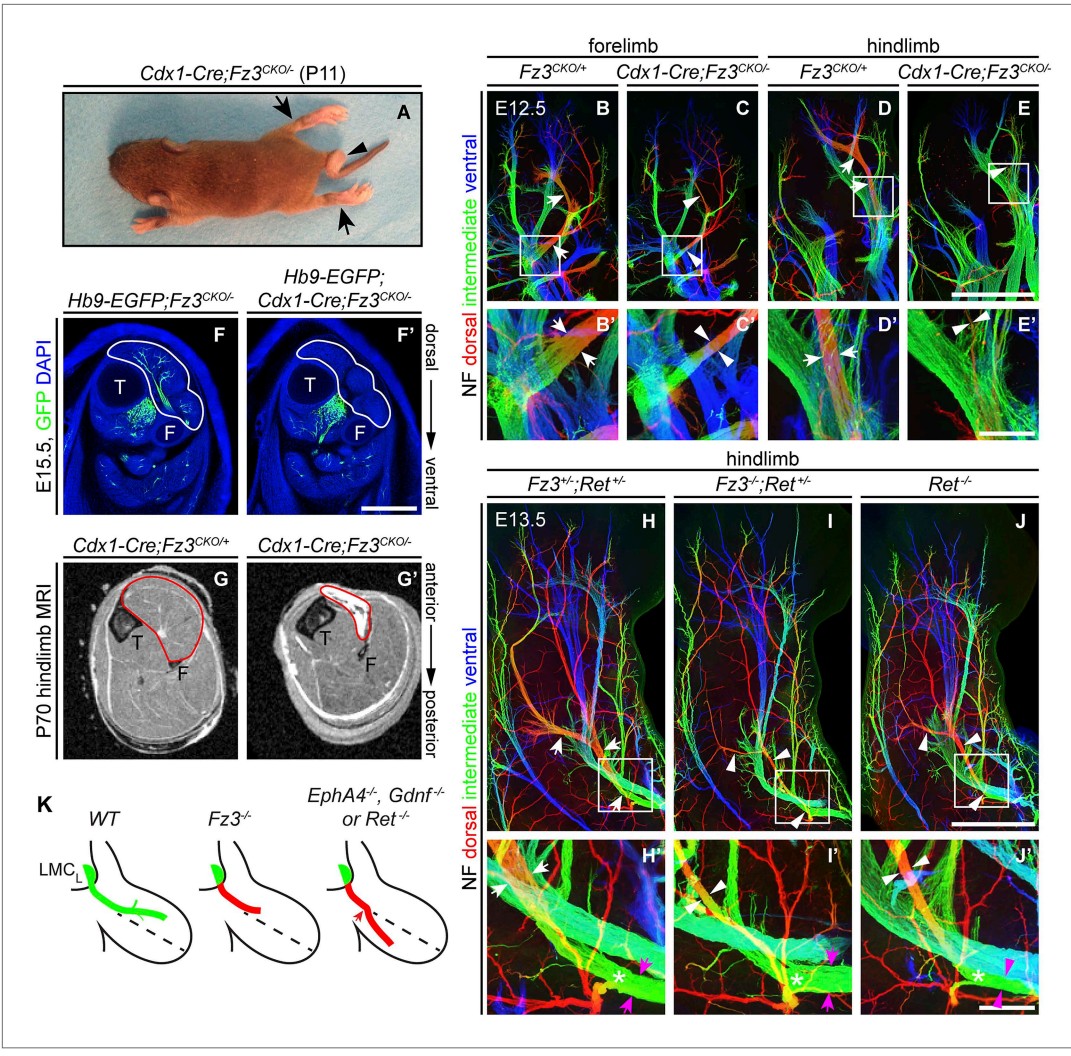

**Figure 7**. Selective hindlimb muscle atrophy in response to the failure of motor innervation in *Fz3*$^{-/-}$ mice. (**A**) P11 *Cdx1-Cre;Fz3*$^{CKO/-}$ mouse exhibiting severe plantar flexion of the hind feet (arrows) and a curled tail (arrowhead). (**B–E'**) NF immunostaining of whole-mount forelimbs and hindlimbs from E12.5 *Fz3*$^{CKO/+}$ and *Cdx1-Cre;Fz3*$^{CKO/-}$ embryos. (**B'–E'**) are magnified views of the boxed regions in (**B–E**). Arrows and arrowheads indicate the dorsal nerve. Depth is color coded as in *Figure 4B–F'*. Scale bars: (**E**), 500 *μm*; (**E'**), 100 *μm*. (**F** and **F'**) In E15.5 *Hb9-EGFP;Cdx1-Cre;Fz3*$^{CKO/-}$ distal hindlimbs, the anterior compartment musculature (tibialis anterior, extensor hallucis longus, and extensor digitorum longus; delimited by a white border) lacks motor innervation as visualized by GFP immunostaining. Extensive motor innervation is seen in the *Hb9-EGFP;Fz3*$^{CKO/-}$ littermate control that lacks the *Cdx1-Cre* transgene. F, fibula; T, tibia. Scale bar, 500 *μm*. (**G** and **G'**) Cross-sectional magnetic resonance images of P70 *Cdx1-Cre;Fz3*$^{CKO/+}$ and *Cdx1-Cre;Fz3*$^{CKO/-}$ hindlimbs show nearly complete degeneration and fibrosis of the anterior compartment musculature in the *Cdx1-Cre;Fz3*$^{CKO/-}$ distal hindlimb. The anterior muscle compartment is delimited by a red border. The large and small dark territories are the tibia and fibula, respectively. (**H–J'**) NF immunostaining of whole-mount hindlimbs from E13.5 *Fz3*$^{+/-}$;*Ret*$^{+/-}$, *Fz3*$^{-/-}$;*Ret*$^{+/-}$, and *Ret*$^{-/-}$ embryos. Boxed regions in (**H–J**) are enlarged in (**H'–J'**). In (**H–J**), white arrows and arrowheads indicate the dorsal nerve. (**H'–J'**) arrow and arrowhead colors: magenta indicates the dorsal nerve diameter proximal and white indicates the dorsal nerve diameter distal to the point where axons stall in *Fz3*$^{-/-}$ limbs (asterisks). Depth is color coded as in *Figure 4B–F'*. Scale bars: (**J**), 500 *μm*; (**J'**), 100 *μm*. (**L**) Comparison of spinal motor axon growth and guidance defects in different knockout lines. In *Fz3*$^{-/-}$ embryos, many dorsal motor axons fail to grow beyond a proximal branch point, whereas in *EphA4*$^{-/-}$, *Gdnf*$^{-/-}$, and *Ret*$^{-/-}$ embryos, dorsal axons are misrouted to the ventral limb.

The following figure supplements are available for figure 7:

**Figure supplement 1**. Comparison of dorsal nerve anatomy in forelimbs with various combinations of *Fz3* and *Ret* loss-of-function alleles.

viscera (sympathetic, parasympathetic, and enteric neurons), DRGs (somatosensory neurons), the adrenal medulla, and a variety of mesenchymal structures in the head (*Dupin et al., 2006*). Genetic defects in melanocyte and enterocyte migration have been extensively studied in mice and humans where they produce easily recognized phenotypes, such as skin depigmentation and aganglionic colon (Waardenberg disease and Hirschprung disease; *Gershon, 2010*). Migration of enteric neurons requires endothelin3/endothelin receptor B and GDNF/Ret signaling. In various experimental contexts, cadherins, integrins, and Eph/Ephrins have also been found to influence NC migration (*McKeown et al., 2013*). Our observation that Fz3 promotes the initial step in NC migration represents a new connection between PCP signaling and NC migration. *Fz3*$^{-/-}$ embryos also exhibit subtle defects in the migration of monoaminergic neurons (*Fenstermaker et al., 2010*) and more severe defects in the migration of VII$^{th}$ cranial nerve neurons (*Qu et al., 2010*), but, in general, neuronal migration defects are not a prominent feature of the *Fz3*$^{-/-}$ phenotype.

At E11.5, a time when only a small fraction of cells within the DRG are dying, the non-migratory NC-derived neuron clusters in *Fz3*$^{-/-}$ embryos show a high density of apoptotic cells. Whether cell death is triggered directly by the failure of cell migration or is a secondary consequence of migratory failure—for example, as a result of incorrect routing of axons produced by neurons within the clusters—it represents an impressive demonstration of the rapidity and efficiency with which the developing nervous system recognizes and eliminates errors. In this respect, it bears a conceptual, and possibly a mechanistic, resemblance to the rapid elimination of the *Fz3*$^{-/-}$ LMC$_L$ neurons with stalled axons.

## Implications of axon stalling and rapid motor neuron death in *Fz3*$^{-/-}$ embryos

Axon stalling has been observed in diverse systems either as part of the normal pattern of axon outgrowth or as a phenotypic consequence of mutation. In the first category are the ~1 day waiting period during which sensory and motor axons stall at the base of the developing chick hindlimb (*Wang and Scott, 2000*) and the alternating phases of stalling and growth that characterize motor axon pathfinding in *C. elegans* (*Knobel et al., 1999*). In the second category are the stalling phenotypes observed among sensory neurons in *Drosophila* embryos carrying mutations in the Neuroglian gene (*Nrg*; *Martin et al., 2008*) or in the gene coding for the PCP protein Flamingo (*Fmi*; the *Drosophila* orthologue of the mammalian *Celsr* genes; *Steinel and Whitington, 2009*), among spinal motor axons in zebrafish carrying collagen XIX mutations (*Beattie et al., 2000*; *Hilario et al., 2010*), and among post-crossing commissural axons in the spinal cord in mouse *Robo1* mutants (*Jaworski et al., 2010*). Although the mechanistic link between extra-cellular signals or adhesive interactions and axon stalling has not been defined for any of these examples, a potentially unifying framework has emerged from studies of the relationship between calcium transients and growth cone motility. Within growth cones, calcium release and motility are inversely correlated, with rapidly moving growth cones showing the lowest frequency calcium transients and stalled or retracting growth cones showing the highest frequency calcium transients (*Gomez and Spitzer, 1999*). In the present study, the loss of *Fz3* blocks growth but does not alter path-finding among dorsal limb motor axons, and therefore presents as a pure stalling phenotype. However, we note that the conceptualization of axon growth and path-finding as distinct behaviors may be simplistic, since the two attributes could be related mechanistically if axon growth is dependent on the same signals that guide the path-finding process.

During normal embryonic development, motor axons compete for limited quantities of survival factors produced by their skeletal muscle targets, with the result that in the prenatal mouse >50% of motor neurons die beginning at E13.5 (*Oppenheim, 1991*; *Sendtner et al., 2000*). Among *Fz3*$^{-/-}$ LMC$_L$ neurons, a wave of cell death begins at E11.5, well before LMC$_L$ axons have reached their final muscle targets, and 2 days before the normal wave of motor neuron cell death (*Kablar and Rudnicki, 1999*; *Oppenheim et al., 2000*). The precocious motor neuron death in *Fz3*$^{-/-}$ embryos is unlikely to reflect survival factors derived from skeletal muscle because even in the complete absence of skeletal myoblasts, myofibers, and muscle—a phenotype produced in *Myf5*$^{-/-}$;*MyoD*$^{-/-}$ embryos—motor neuron cell death does not occur prior to E13.5 (*Kablar and Rudnicki, 1999*). Taken together, the *Fz3*$^{-/-}$ and *Myf5*$^{-/-}$;*MyoD*$^{-/-}$ phenotypes imply that motor neuron survival between E11.5 and E13.5 is an active process and is independent of the motor axon's ultimate target. One class of models that could account for the precocious motor neuron death in *Fz3*$^{-/-}$ embryos invokes the existence of neuronal survival factors produced by intermediate targets along an axon's trajectory, as suggested by in vitro

co-culture experiments with spinal cord commissural axons and floor plate tissue (*Wang and Tessier-Lavigne, 1999*). The *Fz3⁻/⁻* experiments reported here provide in vivo evidence for this model, and suggest that the hypothesized survival factor(s) reside distal to the plexus but proximal to the developing muscle target. We note that alternative models are also consistent with the data, including, for example, the possibility of anti-survival signals at the point of *Fz3⁻/⁻* axon stalling.

While the precocious cell death of *Fz3⁻/⁻* motor neurons can be rescued by eliminating *Bax*, the axon stalling phenotype remains unchanged. This observation is consistent with a model in which a defect in Fz3-dependent axonal growth induces apoptotic cell death, and it argues against an alternate model in which Fz3 functions in some other process to maintain cell viability, with defective axonal growth representing a secondary effect of the neuron's impending demise. A corollary to this argument is that controlling axonal growth may be the only function of Fz3 in motor neurons.

## Planar cell polarity components in axon growth and guidance

In earlier work, *Fz3⁻/⁻*, and *Celsr3⁻/⁻* forebrains were found to have essentially identical defects in multiple axon tracts (*Wang et al., 2002,2006b*; *Tissir et al., 2005*), and in *Fz3⁻/⁻*, *Celsr3⁻/⁻*, and *Vangl2^{Lp/Lp}* spinal cords the commissural axons of dorsal horn sensory neurons were found to exhibit the same rostral turning defect (*Lyuksyutova et al., 2003*; *Tissir et al., 2005*; *Price et al., 2006*; *Shafer et al., 2011*). The present work strongly suggests that, in the context of motor axon growth and guidance, Fz3 and one or more Celsr family members are also likely to act together in the same pathway.

In a wide variety of epithelia, planar cell polarity proteins are arranged in asymmetric cell surface complexes (*Simons and Mlodzik, 2008*; *Goodrich and Strutt, 2011*). At the junctions between adjacent cells, the large cadherin-containing Stan/Fmi/Celsr proteins accumulate at the plasma membranes of both cells and are presumed to form homophilic adhesive interactions. By contrast, Frizzled proteins accumulate at the plasma membrane of one cell and Vang/Stan/Vangl proteins accumulate at the plasma membrane of the opposing cell. The general assumption in the field is that the asymmetric distribution of Frizzled and Vangl proteins is a prerequisite for transmitting polarity information within the plane of the epithelium (*Goodrich and Strutt, 2011*; *Peng and Axelrod, 2012*).

The current conceptual framework appears well suited to explain signal transmission via cell surface complexes of PCP proteins within relatively stable epithelial monolayers. This framework can also accommodate more dynamic situations in which epithelial cells divide or exchange partners on time scales of minutes to tens of minutes, as seen, for example, during convergent extension movements. Although the dynamics of PCP protein complexes remains largely undefined, one study of their trafficking during the cell cycle shows that these complexes can dis-assemble and reassemble on a time scale of tens of minutes (*Devenport et al., 2011*). The axonal development phenotypes produced by loss of Fz3 or Celsr3 raise an apparent paradox if we imagine that Fz3 and Celsr3 sense and integrate environmental information to control axonal growth by forming the same type of cell-surface complexes in growth cones—and, in particular, in growth cone filopodia—as they do in epithelia. Such a scenario would imply that the time scales of assembly and dis-assembly of PCP protein complexes and engagement with and disengagement from partners in the environment would need to occur on the time scale of growth cone dynamics, that is seconds to tens of seconds (*Schaefer et al., 2002*).

At present, the subcellular location of endogenous PCP proteins in developing neurons has not been determined. However, experiments with dissociated commissural neurons that express transfected Fz3 or Vangl2 tagged with fluorescent reporters show that Fz3 is enriched in intracellular vesicles and Vangl2 is localized throughout the growth cone plasma membrane with the highest concentration in filopodia (*Shafer et al., 2011*). While current models of PCP protein action in growth cones are necessarily speculative, the axon growth phenotypes suggest that the signaling repertoire of PCP proteins is more diverse than would be predicted from considerations based only on epithelia. These and other Frizzled and Celsr phenotypes suggest that 'tissue polarity', the original name for the signals sent and received by this pathway, might be more apt than the epithelium-centric name 'planar cell polarity'.

## Materials and methods

### Mouse lines

The *Fz3^{CKO}* allele was generated by homologous recombination in mouse embryonic stem (ES) cells using standard techniques. The targeting construct contained exon 3 with 3.4 kb of 5′ and 6.4 kb of 3′ flanking intronic sequences. An HA epitope was inserted in the linker region that connects the

cysteine-rich ligand-binding domain and the first transmembrane domain (a region encoded in the third exon). A *neomycin phosphotransferase* (*neo*) selection marker with flanking *Frt* sites was inserted 3' of exon 3, and *LoxP* sites were placed 5' of exon 3 and 3' of *neo* (*Figure 6—figure supplement 1A*). The targeting construct was electroporated into R1 mouse ES cells, and colonies were grown in medium containing G418 and ganclovir. Colonies were screened by Southern blot hybridization, and positive clones were injected into C57BL/6 blastocysts to generate chimeric founders. Germline transmission was confirmed by Southern blot hybridization and PCR. The *neo* cassette was removed by crossing to germline *Flp* mice (*Rodriguez et al., 2000*).

The following mouse lines were also used: *Fz3$^{-/-}$* (*Wang et al., 2002*), *Bax$^{-/-}$* (*Knudson et al., 1995*), *R26iAP* (*Badea et al., 2009*), *Hb9-EGFP* (*Wichterle et al., 2002*), *ChAT-IRES-Cre* (JAX #006410; The Jackson Laboratory, Bar Harbor, ME), *Olig2$^{Cre}$* (i.e., a Cre knock-in at the *Olig2* locus; *Dessaud et al., 2007*), *Wnt1-Cre* (*Danielian et al., 1998*), and *Cdx1-Cre* (*Hierholzer and Kemler, 2009*).

## Alkaline phosphatase (AP) histochemistry

Throughout this work, the first day after finding a copulation plug was counted as embryonic day 0.5 (E0.5). For the analysis of E18.5 embryos, skinned heads were immersion fixed in 4% paraformaldehyde (PFA) at 4°C overnight with gentle shaking, washed three times in phosphate buffered saline (PBS), pH 7.4, and then decalcified in 50 mM EDTA, pH 7.0 at 4°C for 1 week. Heads were embedded in 3% low melting point agarose in PBS and sectioned at 200 $\mu$m on a vibratome. AP histochemistry was performed essentially as previously described (*Badea et al., 2003*). Sections were washed twice in PBS with 2 mM MgCl$_2$, heated in a water bath at 69°C for 90 min to inactivate endogenous AP activity, and then equilibrated in AP staining buffer (0.1 M Tris, 0.1 M NaCl, 50 mM MgCl$_2$, pH 9.5). AP histochemistry were carried out in AP staining buffer containing 0.34 $\mu$g/ml 4-nitro blue tetrazolium chloride (NBT) and 0.175 $\mu$g/ml 5-bromo-4-chloro-3-indolyl-phosphate (BCIP) (Roche Applied Science; Indianapolis, IN), for 1 hr to overnight at room temperature with gentle horizontal rotation. Before imaging, sections were dehydrated through an ethanol series and then cleared with BBBA [2:1 benzyl benzoate (BB)/benzyl alcohol (BA)] (Sigma-Aldrich; St. Louis, MO).

## Immunohistochemistry

The following primary antibodies were used: mouse monoclonal anti-Neurofilament (165 kDa) (2H3; Developmental Studies Hybridoma Bank [DSHB]; Iowa City, IA), mouse monoclonal anti-Islet1 (39.3F7; DSHB), mouse monoclonal anti-Islet1/2 (39.4D5; DSHB), rabbit anti-Brn3a (MAB1585; Millipore; Billerica, MA), rabbit anti-Caspase3 (9661; Cell Signaling Technology; Boston, MA), rabbit anti-Foxp1 (ab16645; Abcam; Cambridge, MA), rabbit anti-GFP (A11122; Invitrogen; Grand Island, NY), goat anti-Sox10 (SC-17342; Santa Cruz Biotechnology; Dallas, TX), rabbit anti-Islet1, anti-Islet2, and anti-Islet1/2 (gifts of Dr Tom Jessell and Susan Morton; Columbia University). Secondary antibodies were Alexa Fluor 488, Alexa Fluor 594, or Alexa Fluor 647 conjugated, donkey anti-goat, goat anti-mouse, or goat anti-rabbit IgG antibodies (Invitrogen). Alexa Fluor 647 α-Bungarotoxin (B35450; Invitrogen) was used to label acetylcholine receptors.

Immunostaining was performed on (1) embryos that were either (a) immersion fixed in 4% PFA at 4°C for 1 hr, washed three times in cold PBS, equilibrated in PBS containing 30% sucrose, embedded in optimal cutting temperature (OCT) compound (Sakura Finetek; Torrance, CA), frozen, and sectioned at 14 $\mu$m on a cryostat; or (b) immersion fixed in 4% PFA at 4°C overnight, washed three times in cold PBS, embedded in 3% low melting point agarose, and sectioned at 100 $\mu$m or 700 $\mu$m on a vibratome; and (2) whole-mount embryos and limbs that were immersion fixed in 4% PFA at 4°C for 2 hr and washed three times in cold PBS.

For immunostaining of cryosections, sections were rinsed in PBS three times, blocked in PBST (PBS with 0.3% Triton X-100) containing 5% normal goat or donkey serum (NGS or NDS) at room temperature for 1 hr, and incubated with primary antibodies in PBST containing 5% NGS or NDS at 4°C overnight. Sections were then washed three times in PBS, and incubated with secondary antibodies in PBST containing 5% NGS or NDS at room temperature for 1 hr. Finally sections were washed three times in PBS, and slides were mounted with Fluoromount-G (Southern Biotech; Birmingham, AL).

For immunostaining of 100 $\mu$m-thick vibratome sections, sections were blocked in PBST containing 5% NGS or NDS at room temperature for 1 hr, and incubated with primary antibodies in PBST containing 5% NGS or NDS at 4°C overnight. Sections were then washed five times in PBST, and incubated with

secondary antibodies in PBST containing 5% NGS or NDS at 4°C overnight. Finally sections were washed five times in PBST, and sections were mounted on slides with Fluoromount-G.

For immunostaining of 700 µm-thick vibratome sections or whole-mount embryos and limbs, samples were first incubated in Dent's Bleach (10% $H_2O_2$, 13.3% dimethyl sulfoxide [DMSO], 53.3% methanol) at 4°C for 24 hr, washed in methanol five times, and fixed in Dent's Fix (20% DMSO, 80% methanol) at 4°C overnight. Samples were washed in PBS three times, incubated with primary antibody in blocking solution (20% DMSO, 75% PBST, 5% NGS, 0.025% sodium azide) at room temperature for 5 days to one week with gentle end-over-end rotation, and then washed five times in PBST. Samples were incubated with secondary antibody in blocking solution at room temperature for 2 days with gentle end-over-end rotation and then washed five times in PBST. Finally samples were dehydrated in 50% methanol/PBS and methanol, and cleared in BBBA. Immunofluorescent signals are stable for >1 week at room temperature in BBBA, and stained tissue samples can be stored for >3 months in methanol at 4°C and returned to BBBA with no discernable loss of image quality or signal strength.

## Microscopy and image analysis

Samples processed for AP histochemistry were imaged using a Zeiss Stemi V11 microscope with a color Axiocam CCD in combination with Openlab software. Immunostained samples were imaged using a Zeiss LSM700 confocal microscope with Zen software. Images of vibratome-cut thick samples or whole-mounts were acquired with a 10X air objective at 10 µm intervals in the Z dimension, and the entire Z stack was either collapsed using a maximum intensity projection or color-coded based on depth. Custom built metal and plexiglass embryo holders consisted of a shallow triangular trough (sides: 2 cm × 2 cm × 1 cm; and depths: 1, 2, 3, or 4 mm) into which the BBBA-cleared embryo could be positioned. The trough was then filled with BBBA, and coverslipped. Imaging a BBBA-cleared E11.5 embryo with a 10X objective, a single fluorescent channel, and 10 µm Z-axis steps generates a ~1.5 Gb file, which can be acquired in ~3 hr on a Zeiss LSM700 confocal microscope.

## Quantification of motor neurons, measurement of nerve diameter, and statistical analysis

The number of motor neurons on each section was manually quantified in ImageJ. To measure the thickness of nerves innervating the limb, lines perpendicular to and extending across the nerve were drawn at designated sites in Adobe Illustrator and the length of these lines was determined. Statistical comparisons (student's *t*-test) were performed using GraphPad Prism 5.

## Magnetic resonance imaging (MRI) of mouse limbs

P70 *Cdx1-Cre;Fz3*$^{CKO/−}$ and control mice from the same litter were fixed by cardiac perfusion. Hindlimbs were imaged by MRI as previously described (*Zhang et al., 2008*).

## Acknowledgements

The authors thank Dr Jiangyang Zhang for performing the MRI, and Dario Bonanomi, Hao Chang, David Ginty, Siyi Huang, Artur Kania, Alex Kolodkin, Alisa Mo, Amir Rattner, Shan Sockanathan, Max Tischfield, Hao Wu, and Ye Yan for helpful advice, discussions, and/or comments on the manuscript. The authors are especially grateful to Drs Andreas Hierholzer and Rolf Kemler for sharing their *Cdx1-Cre* mouse line. Supported by the Howard Hughes Medical Institute.

## Additional information

### Competing interests

JN: Reviewing editor, *eLife*. The other authors declare that no competing interests exist.

### Funding

| Funder | Author |
| --- | --- |
| Howard Hughes Medical Institute | Jeremy Nathans |

The funder had no role in study design, data collection and interpretation, or the decision to submit the work for publication.

## Author contributions

ZLH, Conception and design, Acquisition of data, Analysis and interpretation of data, Drafting and revising the article; PMS, Constructed the *Fz3* conditional knockout mouse; JN, Conception and design, Analysis and interpretation of data, Drafting and revising the article

## Ethics

Animal experimentation: This study was performed in strict accordance with the recommendations in the Guide for the Care and Use of Laboratory Animals of the National Institutes of Health. All of the animals were handled according to approved Institutional Animal Care and Use Committee (IACUC) protocol MO11M29 of the Johns Hopkins Medical Institutions.

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
