## [Decision Letter]

Thank you for sending your work entitled “*Frizzled3* controls axonal development in distinct populations of cranial and spinal motor neurons” for consideration at *eLife*. Your article has been favorably evaluated by a Senior editor and 4 reviewers, one of whom is a member of our Board of Reviewing Editors.

The following individuals responsible for the peer review of your submission have agreed to reveal their identity: Robb Krumlauf served as the Reviewing editor and Yimin Zou was one of the peer reviewers.

The Reviewing editor and the other reviewers discussed their comments before we reached this decision, and the Reviewing editor has assembled the following comments to help you prepare a revised submission.

The consensus opinion of the reviewers is that they are enthusiastic about this paper and would like to see a revised version published in *eLife*. It presents an extensive description of novel motor neuron and peripheral nervous system defects in *Fz3* mutants. The genetic experiments are elegant, the experiments thoughtful, and the data striking. There are two primary concerns that need to be addressed and they are summarized below.

1) The first is that the density of the results and the extensive survey of phenotypes while impressive make it challenging for the reader to take away the key findings from the study. The volume of data is at times overwhelming, which makes it hard to work through and appreciate the subtleties of the many observations. Thought needs to be given to finding a coherent way to summarize the key results in an effective way for the reader in the Discussion. Perhaps the authors could emphasize more clearly the role of Frizzled3 in motor neuron target innervation and survival in the periphery, which is indeed the main finding being reported. Peripheral growth and innervation is a very important issue in the context of peripheral nerve injury and peripheral neuropathy. One suggestion is to simplify the data. This paper is filled with detailed results that make it harder for readers for grasp the main findings. Since the motor axon defects are mostly on the dorsal limb, the authors could focus on showing these dorsal axons. Showing other axons that do not display defects makes it very convincing that only a selected group of axons are affected but these might unnecessarily overwhelm the reader. The authors could simply show one to two images of these as “controls” and then clearly illustrate the main phenotype. This will sharpen the focus on the key results.

2) The second general issue relates to postulated mechanisms or level of mechanistic insight underlying the phenotypes. Because of the effort made in describing so many diverse phenotypes the study does not go into much mechanistic characterization that could help to understand how Fz3 acts, why specific motor neuron subtypes are affected and/or a molecular explanation for the axon stalling. As a consequence, the paper presents interpretations or speculation on mechanisms associated with phenotypes, which may have alternative explanations.

For example, the difference between axon outgrowth and guidance needs to be made clearer. The point about outgrowth is emphasized throughout the paper but other interpretations are also possible. The authors show that axons stall and then die but it is premature to conclude based on the limb phenotype alone that axons die because they don't reach an intermediate target. The statement that neurons with long-range axons are programmed to die unless their axons arrive at intermediate targets on schedule is not really supported by the data. How can a role for intermediate targets be distinguished from a role of the final target in survival? It is unclear whether the *Fz3* mutant motor neurons die because of an intrinsic neuronal defect, a defect in the cell's ability to transport survival signals from intermediate or distal targets, or the failure of LMCl axons to contact distal targets that supply trophic support. Integrating these possibilities into the Discussion and less speculation on the mechanism would seem appropriate in the absence of supporting data.

The reviewers are hesitant to request additional experimental detail to complement an already dense study. However, if the report included explant assays it might be possible to determine whether LMCl axons lack an intrinsic ability to extend long axons versus a more direct role of *Fz3* in axon targeting.

---

## [Author Response]

*There are two primary concerns that need to be addressed and they are summarized below*.

*1) The first is that the density of the results and the extensive survey of phenotypes while impressive make it challenging for the reader to take away the key findings from the study. The volume of data is at times overwhelming, which makes it hard to work through and appreciate the subtleties of the many observations. Thought needs to be given to finding a coherent way to summarize the key results in an effective way for the reader in the Discussion. Perhaps the authors could emphasize more clearly the role of Frizzled3 in motor neuron target innervation and survival in the periphery, which is indeed the main finding being reported. Peripheral growth and innervation is a very important issue in the context of peripheral nerve injury and peripheral neuropathy. One suggestion is to simplify the data. This paper is filled with detailed results that make it harder for readers for grasp the main findings. Since the motor axon defects are mostly on the dorsal limb, the authors could focus on showing these dorsal axons. Showing other axons that do not display defects makes it very convincing that only a selected group of axons are affected but these might unnecessarily overwhelm the reader. The authors could simply show one to two images of these as “controls” and then clearly illustrate the main phenotype. This will sharpen the focus on the key results*.

To decrease the amount of data in the regular figures and focus the presentation, we have taken roughly 50% of the panels from Figure 2 (panels Ca-g’, Da-d’, E, E’, and I-K’; 30 panels total) and transferred them to a figure supplement. This moves most of the data regarding cranial nerves and cranial nerve nuclei to the figure supplements. We have also changed panel B’ of Figure 1, which, in the original version, showed a potentially confusing superposition of two nerves from different Z-planes that created the illusion that the two might be connected. Finally, we have focused the writing in the Results section to ensure that each sub-section has a clear statement of the experimental question, the experimental results, and the interpretation of those results.

*2) The second general issue relates to postulated mechanisms or level of mechanistic insight underlying the phenotypes. Because of the effort made in describing so many diverse phenotypes the study does not go into much mechanistic characterization that could help to understand how Fz3 acts, why specific motor neuron subtypes are affected and/or a molecular explanation for the axon stalling. As a consequence, the paper presents interpretations or speculation on mechanisms associated with phenotypes, which may have alternative explanations*.

*For example, the difference between axon outgrowth and guidance needs to be made clearer. The point about outgrowth is emphasized throughout the paper but other interpretations are also possible. The authors show that axons stall and then die but it is premature to conclude based on the limb phenotype alone that axons die because they don't reach an intermediate target. The statement that neurons with long-range axons are programmed to die unless their axons arrive at intermediate targets on schedule is not really supported by the data. How can a role for intermediate targets be distinguished from a role of the final target in survival? It is unclear whether the* Fz3 *mutant motor neurons die because of an intrinsic neuronal defect, a defect in the cell's ability to transport survival signals from intermediate or distal targets, or the failure of LMCl axons to contact distal targets that supply trophic support. Integrating these possibilities into the Discussion and less speculation on the mechanism would seem appropriate in the absence of supporting data*.

We have gone through the text to clarify the writing so that there is a clear distinction between (a) the experimental outcomes and (b) the hypotheses regarding the possible mechanisms underlying those outcomes. In the Discussion, we have also clarified the distinction between axon outgrowth and guidance, although we note that at a mechanistic level these seemingly distinct characteristics may be interconnected, since a failure to grow at a particular point (the phenotype that we observe among *Fz3*^*-/-*^ motor axons that were destined to innervate the dorsal hind limb) may reflect an underlying inability to choose a correct path at that juncture.

We agree with the editor’s comment that “it is premature to conclude based on the limb phenotype alone that axons die because they don't reach an intermediate target”. However, we would still like to propose this as the leading hypothesis to explain the data. Regarding the editor’s next question and sentence: “How can a role for intermediate targets be distinguished from a role of the final target in survival? It is unclear whether the Fz3 mutant motor neurons die because of an intrinsic neuronal defect, a defect in the cell's ability to transport survival signals from intermediate or distal targets, or the failure of LMCl axons to contact distal targets that supply trophic support.” This indicates that we have not clearly explained the argument. Our logic is as follows: (1) in the *Fz3*^*-/-*^ spinal cord, the axons of motor neurons that innervate the dorsal limb stall at the base of the limb at E11.5; (2) the same motor neuron population (and only that population) undergoes massive cell death at the same age (E11.5); (3) in WT embryos, motor axons reach their final targets 2 days later (E13.5), at which time ∼50% of motor neurons start to undergo developmental cell death; and (4), as described in the Discussion, the genetic elimination of all muscle cells (the distal/final target of motor neurons) in *Myf5*^*-/-*^;*MyoD*^*-/-*^ embryos (Kablar and Rudnycki, 1999) produces massive motor neuron death, but this wave of cell death only begins at the normal time, E13.5, i.e., when the motor axons arrive at the place where the muscle targets should be. Points (3) and (4) imply that the time window when trophic support from distal/final muscle targets becomes required for motor neuron survival begins at E13.5, as judged by the timing of motor neuron death in the absence of that support. Points (1) and (2) demonstrate that in *Fz3*^*-/-*^ embryos, axon stalling and precocious motor neuron death occur two days prior to the arrival of WT motor axons at their final targets and two days prior to the time when trophic support from distal/final targets is needed to maintain WT motor neuron viability. Putting together points (1)-(4), we conclude that motor neuron death in *Fz3*^*-/-*^ embryos in not due to trophic effects from the final distal/targets. Furthermore, the coincidence in time between axon stalling and motor neuron death in the same population of motor neurons in *Fz3*^*-/-*^ embryos suggests that the two events may be mechanistically coupled, although a skeptic could argue that loss of *Fz3* produces axon stalling and cell death in the same subset of neurons at the same time via two unrelated processes.

We think that the newly included paragraph in the Discussion section that is devoted to this issue clearly lays out this logic. Here and elsewhere we have been careful to indicate that the intermediate target hypothesis is just a hypothesis, not an established fact. In the Abstract, we characterize the data as “providing in vivo evidence for the idea that developing neurons…” and in the Discussion we characterize the intermediate target hypothesis as a “model” and we write that the data “provide in vivo evidence for this model”, and that “alternative models are also consistent with the data”.

*The reviewers are hesitant to request additional experimental detail to complement an already dense study. However, if the report included explant assays it might be possible to determine whether LMCl axons lack an intrinsic ability to extend long axons versus a more direct role of* Fz3 *in axon targeting*.

We agree that additional experimental data could add substantially to this work – in particular, the identification of putative growth and/or guidance molecules that act through Fz3, or putative intermediate target-derived trophic factors. However, such an undertaking would constitute a major and independent body of work. In vitro culture experiments that merely recapitulate the in vivo behavior of *Fz3*^*-/-*^ axons without leading to real mechanistic insights would add only modestly to the present picture. As the work now stands, we believe it constitutes a comprehensive in vivo body of experimental data with several novel and well-documented biological findings, and therefore it makes sense to publish the body of work in essentially its present form. We have added a paragraph to the Discussion section to provide more background on axon stalling and to address the growth vs guidance issue.

“Axon stalling has been observed in diverse systems either as part of the normal pattern of axon outgrowth or as a phenotypic consequence of mutations…”